# Near-infrared light-triggered prodrug photolysis by one-step energy transfer

Kaiqi Long [1,2,3], Wen Lv[3,4], Zihan Wang[1,2,3], Yaming Zhang [1,2,3], Kang Chen [1,3], Ni Fan[1,2,3], Feiyang Li [5], Yichi Zhang[1,2,3] & Weiping Wang [1,2,3] ✉

Prodrug photolysis enables spatiotemporal control of drug release at the desired lesions. For photoactivated therapy, near-infrared (NIR) light is preferable due to its deep tissue penetration and low phototoxicity. However, most of the photocleavable groups cannot be directly activated by NIR light. Here, we report a upconversion-like process via only one step of energy transfer for NIR light-triggered prodrug photolysis. We utilize a photosensitizer (PS) that can be activated via singlet-triplet (S-T) absorption and achieve photolysis of boron-dipyrromethene (BODIPY)-based prodrugs via triplet-triplet energy transfer. Using the strategy, NIR light can achieve green light-responsive photolysis with a single-photon process. A wide range of drugs and bioactive molecules are designed and demonstrated to be released under low-irradiance NIR light (100 mW/cm², 5 min) with high yields (up to 87%). Moreover, a micellar nanosystem encapsulating both PS and prodrug is developed to demonstrate the practicality of our strategy in normoxia aqueous environment for cancer therapy. This study may advance the development of photocleavable prodrugs and photoresponsive drug delivery systems for photo-activated therapy.

Photolysis, also called as photo-uncaging or photocleavage reaction, has been utilized for controlling molecular functions or release of desired components with light irradiation[1]. It has been identified as a powerful approach with spatiotemporally controllable manner and showed excellent strengths in imaging, photocatalysis, photopharmacology, neuroscience, and drug delivery. Photocleavable prodrugs, with tailor-made structures composed of photoremovable protecting groups (PPGs) and drug molecules, have been developed for light-triggered precise drug activation[2,3]. After systematic administration of photocleavable prodrugs, local light irradiation can be applied onto disease lesions to specifically activate the prodrugs in situ, reducing systemic toxicity and thus increasing biocompatibility and therapeutic efficacy[4,5]. In biomedical applications, near-infrared

(NIR) light (650–900 nm) is highly desirable for photoactivated therapy, due to its deep tissue penetration and low phototoxicity[6,7]. However, the low photon energy of NIR light usually cannot meet the direct activation threshold of the commonly used PPGs, such as PPGs based on nitrobenzene[8], coumarin[9–11] and boron-dipyrromethene (BODIPY)[12].

There have been strategies to achieve long-wavelength light-triggered prodrug photolysis, including 1). Increasing the absorption wavelength through molecular modifications; 2). Two-photon excitation; 3). Photon upconversion systems. Usually, the absorption wavelength can be increased by expanding its π conjugation or substitution of chemical groups. However, it was time and labor consuming, while the chemical modifications on photocages may also affect their

[1]State Key Laboratory of Pharmaceutical Biotechnology, The University of Hong Kong, Pokfulam, Hong Kong, China. [2]Department of Pharmacology & Pharmacy, Li Ka Shing Faculty of Medicine, The University of Hong Kong, Pokfulam, Hong Kong, China. [3]Laboratory of Molecular Engineering and Nanomedicine, Dr. Li Dak-Sum Research Centre, The University of Hong Kong, Pokfulam, Hong Kong, China. [4]State Key Laboratory of Organic Electronics and Information Displays, Jiangsu Key Laboratory for Biosensors, Institute of Advanced Materials (IAM), Nanjing University of Posts & Telecommunications, Nanjing, China. [5]School of Environmental and Chemical Engineering, Jiangsu University of Science and Technology, Zhenjiang, China. ✉e-mail: wangwp@hku.hk

photolysis efficiency[13,14]. Moreover, the two-photon excitation of photocages requires femtosecond pump laser with high irradiances ($10^6$ W/cm² or above) and the reaction can only occur at the laser focal point[15]. Lanthanide-doped upconversion nanoparticles (UCNPs) have emerged as reliable platforms for turning NIR light into UV/visible light, thus enabling long-wavelength light to activate short-wavelength light-responsive PPGs[16,17]. However, the required excitation irradiance is still relatively high ($10^1$–$10^4$ W/cm²), since UCNPs exhibit low absorption coefficient and cross-sections, and the efficiency of luminescence resonance energy transfer (LRET) between UCNP and PPGs remains unsatisfactory. Triplet-triplet annihilation-based upconversion (TTA-UC) is another upconversion strategy for long-wavelength light-triggered photolysis, which depends on multi-step energy transfer between photosensitizer (PS) and annihilator to produce upconverted photons (Fig. 1a)[18,19]. TTA-UC enabled the utilization of low-irradiance long-wavelength light ($10^{-3}$–$10^{-1}$ W/cm²), however, the internal energy consumption during the multi-step energy transfer processes still resulted in low quantum yields and photolysis efficiency.

To overcome these concerns, we previously developed an upconversion-like strategy by sensitizing the prodrug to its triplet excited state with a single low-energy photon (Fig. 1b)[20,21]. As the mechanism, long-wavelength light activates PS from ground state ($S_0$) to singlet excited state ($S_1$), followed by intersystem crossing (ISC) to its triplet excited state ($T_1$) and the activation of prodrug through triplet-triplet energy transfer. This process is like the photon upconversion, but only a single photon is involved. So far, the anti-Stocks shift of this process is still limited (that is, we only can use red light to activate green light-responsive photolysis), since the photon energy needs to be higher than the $S_1$ state of PS, and only the PS with low singlet state energy level can be used (e.g., platinum- or palladium-coordinated porphyrins).

In this study, we develop a strategy for efficient prodrug photolysis that overcomes the limitation of photon energy which must be higher than the $S_1$ state of PS by utilizing PS with singlet-to-triplet (S-T) absorption (denoted as STPS). Since STPS can be directly activated to $T_1$ state from $S_0$ state, the photon with energy higher than $T_1$ of STPS is enough to initiate the upconversion-like photolysis, reducing the excitation energy level to achieve NIR light activation (Fig. 1c). Multiple steps of energy transfer can be bypassed, resulting in less internal energy loss and higher photolysis efficiency. In the presence of STPS, a series of short-wavelength light-responsive photocage-conjugated prodrugs, including the prodrugs of chlorambucil (Cb), vadimezan, indomethacin, naproxen, ibuprofen, benzyloxycinnamic acid, tetracaine, dopamine, tyramine, and homoveratrylamine, are activated by NIR light with low irradiance and short duration (100 mW/cm², 5 min) at high yields (up to 87%). Surprisingly, the yields of drug release are even higher than that directly triggered by short-wavelength light. This strategy is verified in vivo by using a micellar nanosystem for light-triggered drug release and photoactivatable cancer therapy. Overall, this study provides a strategy to utilize the low-energy photons of long-wavelength light to trigger prodrug photolysis with high yields, which demonstrates the potential for advanced photoactivatable therapy.

## Results

### Preparation and characterizations of Osmium-based STPS and BODIPY-Cb prodrug

We have developed a BODIPY-Cb prodrug that can release Cb as an anti-cancer agent for photoactivated therapy under green-light irradiation[20]. As STPS, Osmium (II) complexes exhibit strong singlet-to-triplet absorption, which enables direct excitation from its $S_0$ state to $T_1$ state[22]. Here, an osmium-containing PS, Os (II) bromophenyl

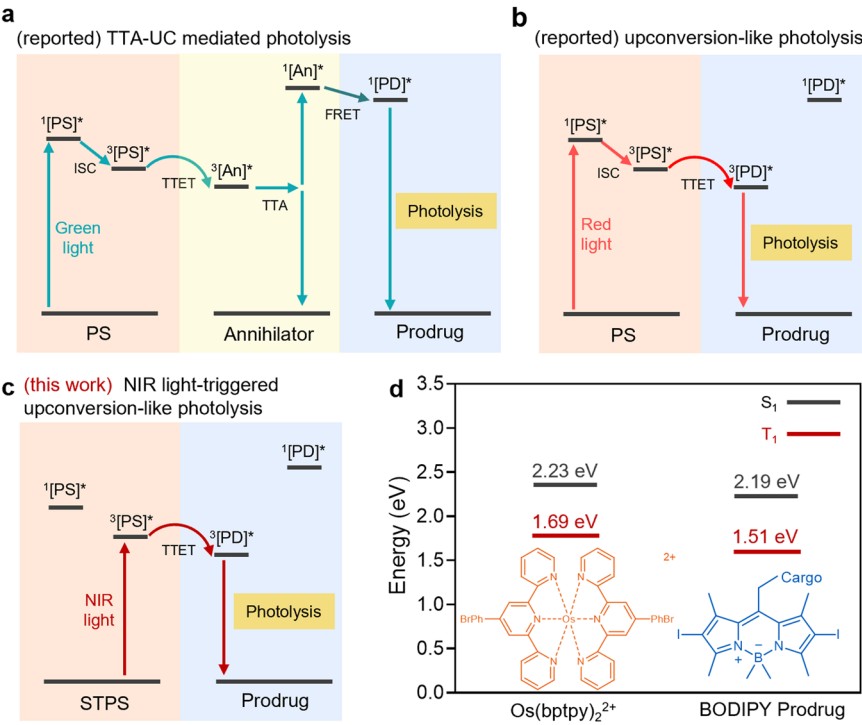

**Fig. 1 | Schematic illustration of reported photolysis strategies and NIR light-triggered photolysis by one-step energy transfer in this work. a** The reported mechanism of TTA-UC-mediated photolysis. **b** The reported mechanism of red light-triggered upconversion-like photolysis. **c** The mechanism of NIR light-triggered photolysis by one-step energy transfer. **d** $S_1$ and $T_1$ energy levels and chemical structures of Os(bptpy)$_2$²⁺ and BODIPY prodrug. TTA-UC triplet-triplet annihilation-based upconversion, PS photosensitizer, An annihilator, PD prodrug, FRET fluorescence resonance energy transfer, TTET triplet-triplet energy transfer, STPS photosensitizer with singlet-to-triplet absorption.

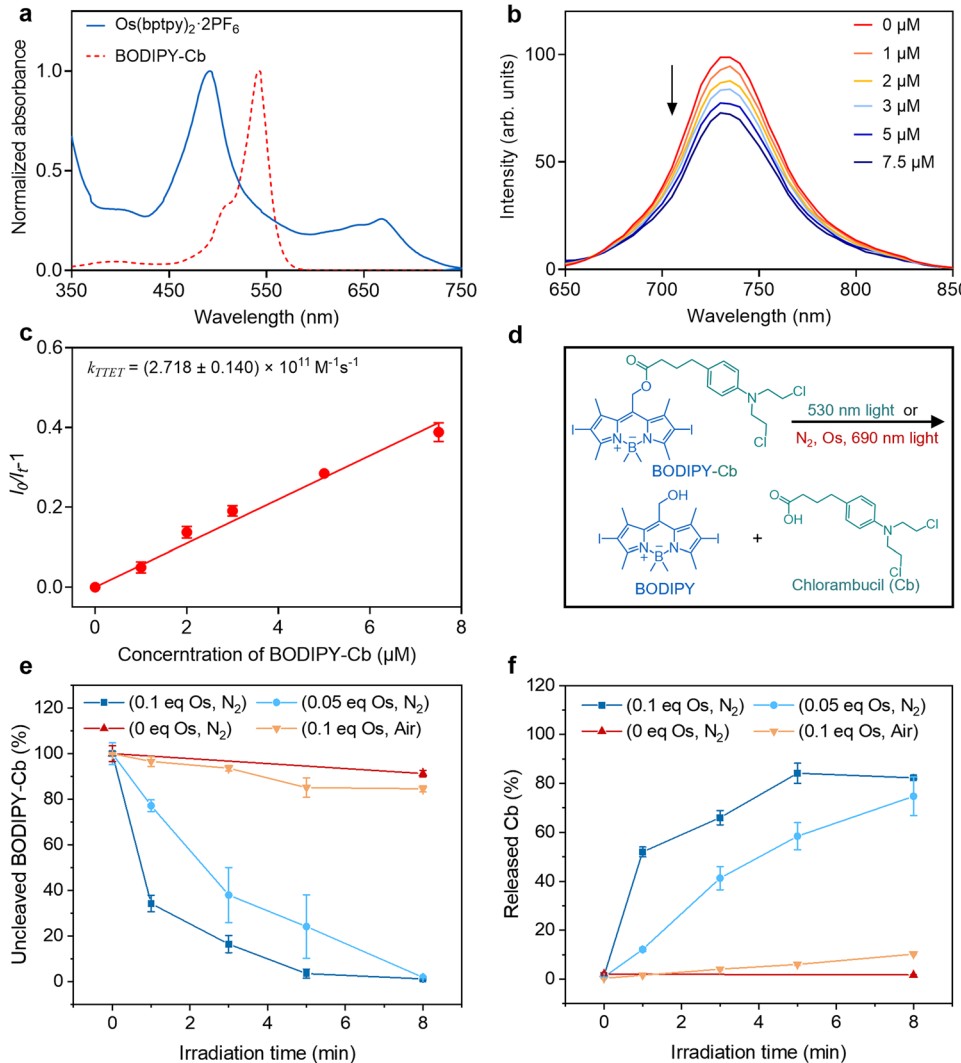

**Fig. 2 | Photophysical properties and photolytic reaction upon light irradiation. a** Normalized absorption spectra of BODIPY-Cb and Os(bptpy)$_2$$^{2+}$ in dichloromethane. **b** Phosphorescence quenching of Os(bptpy)$_2$$^{2+}$ (10 μM) in the presence of different concentrations of BODIPY-Cb, $\lambda_{ex}$ = 492 nm, in N$_2$-saturated toluene. **c** Stern-Volmer plot based on the phosphorescence quenching by different concentrations of BODIPY-Cb. **d** Photolysis reaction of BODIPY-Cb with/without Os(bptpy)$_2$$^{2+}$ (denoted as Os). **e** Photolysis rate of BODIPY-Cb upon 690 nm light irradiation for different time durations. **f** Generation rate of Cb upon 690 nm light irradiation for different time durations (solution: N$_2$-saturated methanol; BODIPY-Cb concentration: 10$^{-3}$ M; light irradiance: 690 nm, 100 mW/cm$^2$). Data in c, e, f are presented as mean ± SD, $n$ = 3 independent experiments. Source data are provided as a Source Data file.

terpyridine complex (Os(bptpy)$_2$$^{2+}$), was synthesized and characterized (referred to the *Supplementary Information*, Supplementary Fig. 2).

As shown in the UV-Vis spectrum, the BODIPY-Cb prodrug exhibits a single absorption peak in the visible-light area (peaked at 543 nm, $\varepsilon$ = 8.94 × 10$^4$ M$^{-1}$ cm$^{-1}$), indicating green light excitation. The Os(bptpy)$_2$$^{2+}$ exhibits broad peaks in the visible-to-NIR area (Fig. 2a). The absorption peak at about 492 nm is identified as the singlet metal-to-ligand charge transfer ($^1$MLCT) absorption, also termed as singlet-singlet (S-S) absorption. Notably, the peak in the far red-NIR area (peaked at 688 nm, $\varepsilon$ = 2.66 × 10$^4$ M$^{-1}$ cm$^{-1}$), is identified as the triplet metal-to-ligand charge transfer ($^3$MLCT) absorption (also termed as S-T absorption), demonstrating that it can be excited by NIR light. The two twistable bromophenyl groups of the bromophenyl terpyridine ligands extend the S-T absorption from 650 nm to 750 nm with increased absorption coefficient. More detailed photophysical properties of BODIPY-Cb and Os(bptpy)$_2$$^{2+}$ were recorded in Supplementary Table 1 in the *Supplementary Information*.

Based on the phosphorescence emission, the T$_1$ energy level of Os(bptpy)$_2$$^{2+}$ was determined as 1.69 eV, which is close to the result of time-dependent density functional theory (TD-DFT) calculation (Supplementary Fig. 5 and Supplementary Table 1)[23]. The theoretical excitation energies of Os agree well with the experimental spectra in Fig. 2a. Notably, the S$_0$-T$_1$ transition was calculated to be at 678 nm, which fitted the experimental results and verified the S-T transition compatibility of Os. The T$_1$ energy level of BC was calculated as 1.51 eV based on the TD-DFT calculations (Supplementary Fig. 6). Moreover, the T$_1$ energy of the BODIPY-OH photocage was determined as 1.54 eV, which is close to that of BC prodrug, implying that T$_1$ energy level of the prodrug mainly depends on its photocage moiety (Supplementary Fig. 7). Based on the energy levels, the molecule pair, Os(bptpy)$_2$$^{2+}$ and BODIPY-Cb, satisfies the energy requirement for triplet-triplet energy transfer (TTET) (T$_1$ (PS)>T$_1$ (PD)) and the upconversion-like process (T$_1$ (PS)<S$_1$ (PD)) (Fig. 1d). After applying NIR light and activating Os(bptpy)$_2$$^{2+}$ to the triplet state (1.69 eV), the photon energy can be transferred to the triplet state of BODIPY prodrug (1.51 eV) and trigger its photolysis.

## TTET between Os(bptpy)$_2^{2+}$ and BODIPY-Cb prodrug

Stern-Volmer phosphorescence quenching assay was used to verify the TTET from Os(bptpy)$_2^{2+}$ to BODIPY-Cb. The phosphorescence of Os(bptpy)$_2^{2+}$ was found to be quenched by titrating BODIPY-Cb prodrug into its N$_2$-saturated toluene solution. It was observed that the phosphorescence of Os(bptpy)$_2^{2+}$ decreased while increasing the BODIPY-Cb concentration, which verifies the energy transfer from $^3$Os(bptpy)$_2^{2+*}$ to $^3$BODIPY1-Cb$^*$(Fig. 2b, c). The TTET rate constant ($k_{TTET}$) was calculated as $(2.72 \pm 0.14) \times 10^{11}\,M^{-1}\,s^{-1}$, indicating efficient energy transfer from T$_1$ state of Os(bptpy)$_2^{2+}$ to T$_1$ state of BODIPY-Cb.

## Photolysis of BODIPY-based prodrugs via upconversion-like process

The photolysis of BODIPY-Cb (Fig. 2d) was studied by irradiating solutions with 530 nm or 690 nm lamps and analyzing the products by high-performance liquid chromatography (HPLC). It was observed that BODIPY-Cb can be cleaved under 530 nm green light, since the high-energy photons can directly activate the prodrug to the S$_1$ state of BODIPY-Cb, followed by the cleavage relaxation and generation of free drug (Supplementary Fig. 8). Notably, the photocleavage was retarded in air-saturated solution, indicated that the cleavage relaxation can occur from the T$_1$ state which was quenched by oxygen. In the existence of Os(bptpy)$_2^{2+}$, both decomposition of the prodrug and generation of free Cb were observed upon 690 nm light irradiation in N$_2$-saturated solution (Fig. 2e, f, and Supplementary Fig. 9). It can be explained by the TTET process, where BODIPY-Cb was promoted to T$_1$ state after accepting the energy from T$_1$ of Os(bptpy)$_2^{2+}$. Also, the generation of free Cb was accelerated while increasing the molar ratio of Os(bptpy)$_2^{2+}$ in the solution. In the existence of 0.1 equiv. of Os(bptpy)$_2^{2+}$, BODIPY-Cb decomposed completely ($96.74 \pm 1.26\%$) upon the light irradiation at 100 mW/cm$^2$ for 5 min and generated free Cb at a relatively high yield of $84.17 \pm 4.21\%$. It should be noted that the yield of free drug is much higher than that of BODIPY-Cb photolysis with shorter wavelength light, including the direct photolysis by green-light irradiation (max. yield at 31.71%) (Supplementary Figure 8) and the upconversion-like photolysis with platinum PS by red-light irradiation (max. yield at 41.74%)[20]. It was reported that the stability of energy acceptors affected the efficiency of upconversion or photochemical reactions[24,25]. Thus, we evaluated the photodamage of the BODIPY-OH photocage under 530 nm green light, 625 nm red light (in the presence of PtTPBP), and 690 nm NIR light (in the presence of Os(bptpy)$_2^{2+}$) (100 mW/cm$^2$, 0-7 min). As shown in Supplementary Fig. 10, the photocage bleached fastest under green light, slowly under red light, and slowest under NIR light, showcasing that utilizing low-energy NIR photons and simplifying energy transfer processes can reduce photodamage of the prodrug and unexpected relaxation of the excited states. High photolysis yield of the free drug may lead to better therapeutic efficacy of the prodrug upon light irradiation. Besides, as expected, the decomposition of BODIPY-Cb, as well as the generation of free Cb, were mostly retarded in the air-saturated solution since the T$_1$ states were quenched by the oxygen molecules (Fig. 2e, f). For details, the quantum yield of photolysis ($\Phi_p$), the quantum yield of drug release ($\Phi_r$) and the cross sections are recorded in Supplementary Table 3.

Prodrug photolysis has emerged as a spatiotemporally controllable process for optochemical control of biological processes. The related studies largely depend on the precise deprotection of bioactive molecules under light illumination[26–28]. Encouraged by the NIR light-triggered photolysis of BODIPY-Cb, we expanded this concept as NIR light-triggered release of different bioactive molecules, including a) anti-cancer drugs, e.g., Cb and vadimezan (DMXAA); b) anti-inflammation drugs, e.g., indomethacin (IDM), naproxen (NPX), ibuprofen (IBF), and benzyloxy cinnamic acid (BCA); c) anesthesia agent, e.g., tetracaine (TCI); d) biogenic amines, e.g., dopamine (DPA), tyramine (TyrA) and homoveratrylamine (HVA) (Fig. 3). BODIPY photocage was conjugated with the free drug molecules to form photoactivatable prodrugs with photolabile linkages, including ester and carbamate. The light illumination was performed with a NIR lamp, of which the parameters were 100 mW/cm$^2$, 690 nm for 5 min, in a N$_2$-saturated solution containing 0.1 equiv. of Os(bptpy)$_2^{2+}$ and 1 equiv. of prodrugs. For BODIPY-Cb (compound 4), the photolytic yield was measured as $84.17 \pm 4.21\%$, as mentioned above. For BODIPY-vadimezan (BODIPY-DMXAA, compound 5), the prodrug was completely consumed with a yield of free DMXAA at $82.52 \pm 7.22\%$ (Supplementary Figure 11). For BODIPY-indomethacin (BODIPY-IDM, compound 6), BODIPY-naproxen (BODIPY-NPX, compound 7), BODIPY-ibuprofen (BODIPY-IBF, compound 8) and BODIPY-benzyloxycinnamic acid (BODIPY-BCA, compound 9), the photolytic yields were $68.95 \pm 4.69\%$, $84.32 \pm 5.79\%$, $48.49 \pm 4.69\%$ and $87.12 \pm 3.17\%$, respectively (Supplementary Figs. 12–15). It should be noted that the above prodrugs (compound 4-9) were fabricated with photolabile ester bonds by conjugating BODIPY photocage and the drug molecules with carboxylic groups. In addition, the photocage was conjugated with drug molecules with amino groups to produce prodrugs with photolabile carbamate bonds. As a result, BODIPY-tetracaine (BODIPY-TCI, compound 10) exhibited photolytic yield at $41.34 \pm 4.61\%$ after 5-min NIR-light irradiation (Supplementary Fig. 16). For BODIPY-dopamine (BODIPY-DPA, compound 11), BODIPY-tyramine (BODIPY-TyrA, compound 12) and BODIPY-homoveratrylamine (BODIPY-HVA, compound 13), the photolytic yields were $69.04 \pm 1.99\%$, $46.57 \pm 2.37\%$, and $74.17 \pm 3.97\%$, respectively (Supplementary Figs. 17–19). Furthermore, BODIPY2-Cb prodrug (compound 14), whose structure is similar to that of BODIPY-Cb but without iodine insertion at the 2- and 6-positions, showed no photolytic yield upon NIR light in the present of Os(bptpy)$_2^{2+}$ (Supplementary Fig. 20), which can be explained by that the lack of iodine atoms leads to insufficient population and fast decay of T$_1$ state[29]. Moreover, the nanosecond transient absorption spectra (Supplementary Fig. 21) and its decay trace (Supplementary Fig. 22) at 540 nm revealed longer T$_1$ lifetime of BODIPY-Cb (compound 4) than that of BODIPY2-Cb (compound 14). In summary, the quantum yields and cross sections of different prodrugs are measured and listed in Supplementary Table 4.

## Photoactivatable nanosystem for NIR light-triggered drug release

NIR light-triggered prodrug activation was then investigated in biological systems for medical applications. Considering that both the prodrug and PS are hydrophobic and the oxygen-sensitive energy transfer process, activating photolabile prodrugs through the upconversion-like mechanism in normoxia aqueous solutions would be challenging. Therefore, we loaded the prodrug and PS in polymeric nanoparticles, which can protect the triplet states from oxygen quenching and thus enable the TTET-based photolysis and drug release in biological environments. An FDA-approved block copolymer, poly(lactic acid)-poly(ethylene glycol) (PLA$_{5000}$-mPEG$_{5000}$), was used to fabricate biocompatible and biodegradable nanoparticles (Fig. 4a)[30]. The ratios of PS (Os) and prodrug (BC) were optimized by feeding different amounts of cargos and recording the size, PDI, and photolytic yields of the prodrug (Supplementary Fig. 23). As shown in Supplementary Table 5, the optimized formulation of Os/BC NPs has the feeding ratio of 0.75% Os (w/w) and 0.5% BC (w/w), and the encapsulation efficiencies are 29.43% and 55.79%, respectively. Dynamic light scattering (DLS) recorded the size of blank nanoparticles (blank NPs), Os(bptpy)$_2^{2+}$-loaded nanoparticles (Os NPs), BODIPY-Cb-loaded nanoparticles (BC NPs), and Os(bptpy)$_2^{2+}$ plus BODIPY-Cb-loaded nanoparticles (Os/BC NPs) at around 50 nm (Supplementary Table 6), which was also verified by TEM imaging (Fig. 4b, c). Excellent colloidal stability of the nanoparticles was observed, of which the size remained stable for at least 72 h at 37 °C (Supplementary Fig. 24). Besides, the absorption spectra of Os/BC NPs

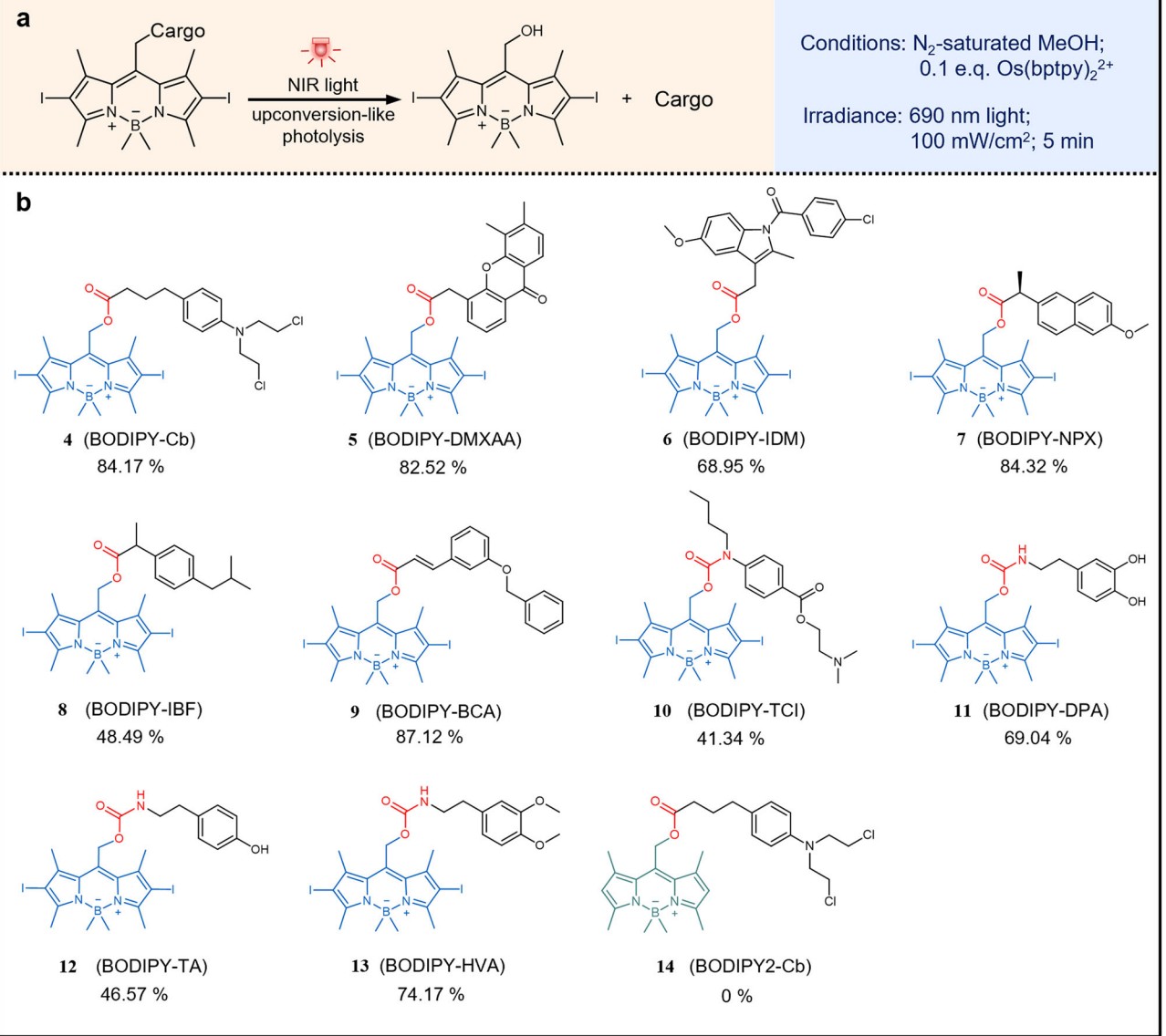

**Fig. 3 | Photolysis reactions of different prodrugs in the presence of Os(bptpy)₂²⁺ and 690 nm light irradiation. a** The reaction and conditions of the prodrug photolysis reaction. **b** Chemical structures of the BODIPY prodrugs. The percentage number represents the photolytic yield of free drug from the respective prodrug, quantitively determined by HPLC.

displayed peaks in both visible area (540 nm) and NIR area (690 nm), indicated successful encapsulation of Os(bptpy)₂²⁺ and BODIPY-Cb in the nanoparticles (Fig. 4d).

Light-triggered drug release has attracted many interests in drug delivery and precise disease treatment[31–33]. Slow release of BC and Os from the nanoparticles was verified (Supplementary Figure 25), which corresponds to the previously reported PLA-PEG-based prodrug nanoparticles[4]. Then, the prodrug activation in Os/BC NPs were investigated under 690 nm light irradiation at 100 mW/cm². The release of free drug Cb as well as the consumption of BODIPY-Cb were obvious after light irradiation, indicating that the upconversion-like photolysis process took place in the Os/BC NPs that were dispersed in normoxia aqueous solutions. Quantitatively, both the decomposition of BODIPY-Cb and release of Cb increased along with the irradiation time from 0 to 30 min (Fig. 4e). The drug release percentage was detected as 62.24% after 30-min light irradiation, while 79.52% of BODIPY-Cb was consumed. As compared, the nanoparticles encapsulating only BODIPY-Cb (BC NPs) showed negligible Cb release upon NIR-light irradiation, indicating the prodrug photolysis within the nanoparticles depends on the

energy transfer from Os(bptpy)₂²⁺ to BODIPY-Cb (Fig. 4f). In all, these observations confirm that Os/BC NPs enabled NIR light-triggered drug release in normoxia aqueous solutions.

### In vitro and in vivo photoactivation of prodrugs for therapy

In vitro and in vivo studies of light-controllable cancer treatment with Os/BC NPs were further conducted. Cb is an FDA-approved anti-tumor drug that has been applied in clinical cancer therapy since 1950s[34]. Some cb prodrugs have been developed, which reduced its systemic side effects by hindering the off-target toxicity and enhanced therapeutic efficacy by precise activation at lesions[35,36]. Here, the cytotoxicity of the light-triggered cb release from the nanoparticles was investigated through the 3-(4,5-dimethylthiazol-2-yl) 2,5-diphenyl tetrazolium bromide (MTT) assay against human cervical cancer (HeLa) cells. It was observed that Os/BC NPs exhibited significant antiproliferation effects towards HeLa cells upon NIR-light irradiation (690 nm, 100 mW/cm², 30 min), while BC NPs displayed negligible toxicity. Os NPs slightly inhibited cell growth after NIR-light exposure, indicating limited phototoxicity without prodrug activation. Besides,

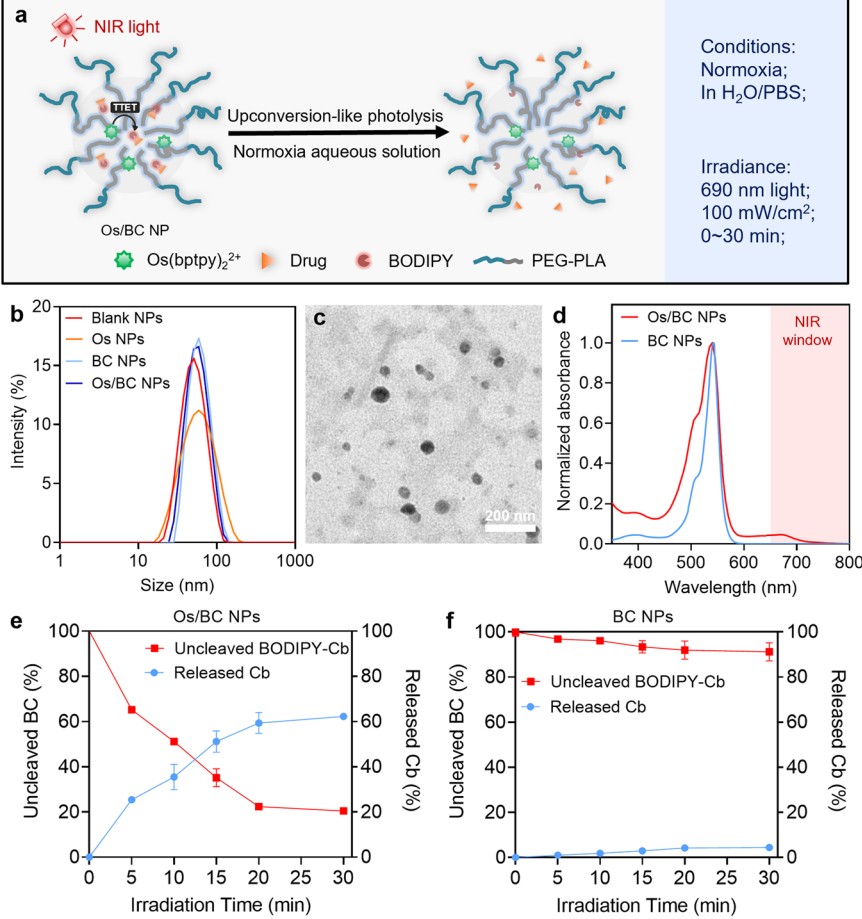

**Fig. 4 | Photoactivatable nanosystem for NIR light-triggered drug release.**
**a** Schematic illustration of NIR light-triggered drug release from the nanoparticle containing Os and BC (Os/BC NP). **b** Size distribution of various nanoparticles.
**c** TEM image of Os/BC NPs. The test was independently repeated in triplicates with similar results. **d** Normalized absorption of BC NPs and Os/BC NPs in aqueous solutions. The red area indicates the NIR window. **e** Percentage of uncleaved BODIPY-Cb and released Cb from Os/BC NPs upon 690 nm light irradiation for 0−30 min. **f** Percentage of uncleaved BODIPY-Cb and released Cb from BC NPs upon 690 nm light irradiation for 0−30 min. Data are presented as mean ± SD, $n$ = 3 independent experiments. Source data are provided as a Source Data file.

in a dark environment, Os NPs, BC NPs, and Os/BC NPs all exhibited minimal cytotoxicity (Fig. 5a–c). The low phototoxicity of Os PS alone can be explained by the limited singlet oxygen generation ability as compared to methyl blue (Supplementary Fig. 26). For further confirmation, live-dead staining analysis was conducted by Calcein AM/PI co-staining assay. Large proportion of dead cells presenting red fluorescence were observed in the Os/BC NPs plus light-treated group, while other groups did not cause obvious cell death (Fig. 5d). The results coincide well with the cytotoxicity study, demonstrating that the light-triggered prodrug activation and drug release from Os/BC NPs efficiently inhibited the growth of cancer cells.

For the other prodrugs, such as BODIPY-indomethacin (BI) and BODIPY-naproxen (BN) (prodrug 6 and 7), similar nanoparticles (Os/BI NPs and Os/BN NPs) were prepared, and their anti-inflammation effects were evaluated by Griess reagent in lipopolysaccharide (LPS)-activated RAW264.7 macrophages. Nitric oxide (NO) is an inflammatory mediator, of which the content reveals the progression of inflammation[37]. Increased NO production was observed after LPS-mediated activation, which was dose-dependently suppressed by Os/BI NPs or Os/BN NPs with light irradiation (Supplementary Fig. 27). These results verified the anti-inflammatory effects of light-triggered release of clinical anti-inflammatory drugs, indomethacin, and naproxen, with negligible cytotoxicity.

To further investigate the mechanisms of cell death triggered by prodrug photolysis, HeLa cells treated with Os NPs, BC NPs, and Os/BC

NPs followed by NIR-light irradiation were stained with Annexin-V FITC/PI to investigate apoptosis process (Fig. 5e). The results showed that about 63.33% of cells were apoptotic after the treatment with Os/BC NPs (equivalent concentration of BC at 10 μM) and NIR-light irradiation, which was dominated by late apoptosis (59.48%). Less proportion of apoptotic cells were observed in the groups treated with Os NPs and BC NPs. In all, the results confirmed that the cytotoxicity of Os/BC NPs was based on the apoptosis-inducing effect after light-triggered release of Cb.

It was reported that PLA-mPEG micellar nanoparticles displayed circulation stability and tumor accumulation ability based on enhanced permeability and retention (EPR) effect in tumors after systemic administration[38,39]. For verification, we labeled the nanoparticles with 1,1′-dioctadecyl-3,3,3′,3′-tetramethylindotricarbocyanine iodide (DiR) dye and examined the biodistribution of DiR-loaded nanoparticles (DiR NPs) after intravenous injection into HeLa tumor-bearing mice. Based on fluorescence observation with an in vivo imaging system (IVIS), it was found that the nanoparticles exhibited both longer circulation time and tumor-accumulation ability as compared to free DiR (Fig. 6a). The fluorescence signal representing DiR NPs was obviously enhanced in tumor areas with the increase of time from 0 h to 24 h, while the free dye was metabolized within the first 8 h. The tumors and major organs were excised for ex vivo fluorescence imaging 24-h post injection. As a result, the nanoparticles exhibited preferential accumulation and retention capability in tumors (Fig. 6b and

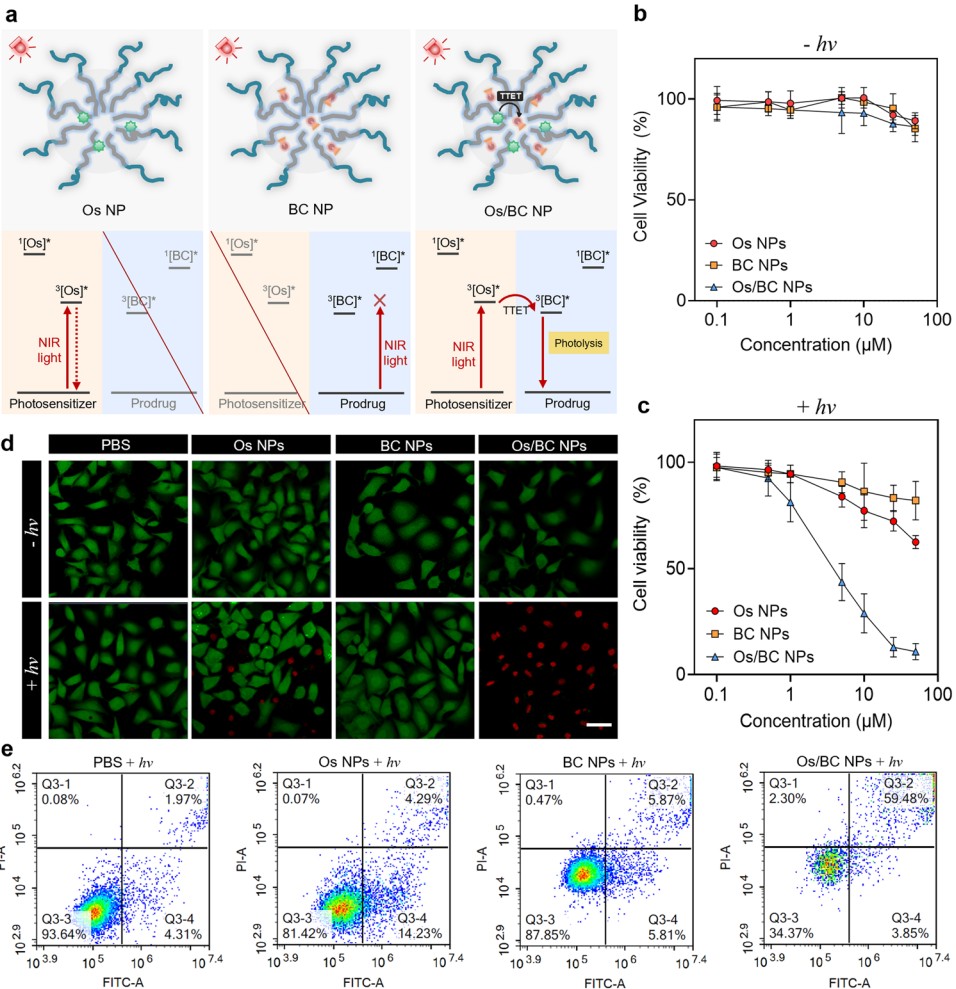

**Fig. 5 | In vitro cytotoxicity with photoactivatable prodrug activation.**
**a** Schematic illustration of Os NP, BC NP, and Os/BC NP, and their Jablonski diagrams of energy transfer processes upon NIR-light irradiation. **b**, **c** Cytotoxicity of Os NPs, BC NPs, and Os/BC NPs with/without light irradiation against HeLa cells. Data are presented as mean ± SD, $n = 3$ independent experiments. **d** Calcein-AM/PI staining of HeLa cells after treatment with Os NPs, BC NPs, and Os/BC NPs with/without light irradiation. Scale bar: 20 μm. **e** Apoptosis study of HeLa cells treated with Os NPs, BC NPs, and Os/BC NPs with light irradiation. Light irradiation: 690 nm, 100 mW/cm², 30 min. Source data are provided as a Source Data file.

Supplementary Fig. 28). It should be noted that the prodrug can be selectively activated in tumors by NIR light while those nanoparticles in the normal tissues and organs will not be activated, which can alleviate the side effects of chemotherapy.

Encouraged by the anti-proliferation effect and tumor retention capability, Os/BC NP was considered as an applicable agent for light-triggered drug release and photoactivated cancer chemotherapy. The HeLa tumor-bearing mice were randomly divided into six groups (n = 5 mice) when the tumors reached at around 100 cm³. PBS, free Cb (8 mg/kg), Os NPs (at the equivalent concentration of Os in Os/BC NPs), BC NPs (at the equivalent concentration of Cb) and Os/BC NPs (at the equivalent concentration of Cb) were intravenously injected on Day 1 (Fig. 6c). At 24 h post injection, NIR-light irradiation (690 nm, 300 mW/cm², 10 min) was applied topically onto the tumor area (Fig. 6d). LC/MS/MS was used to quantify the in vivo prodrug activation. It was found that both BC and Cb exhibited obvious relative abundance in LC/MS/MS chromatograms at low concentrations (1-400 ng/mL and 1-100 ng/mL, respectively)) (Supplementary Figs. 29 and 30). At 24 h after the i.v. injection of OS/BC NPs, it was observed that the BC prodrug was distributed mainly in tumor tissues and the major organs (Fig. 6e). Obvious BC consumption was observed in tumors after NIR light irradiation, as well as the release of free Cb (Fig. 6f). Notably, since the light irradiation was only performed at the tumor area, no free Cb was observed in the major organs, indicating the excellent tumor specificity of the treatment.

For evaluating the therapeutic efficacy, formulation injection and light irradiation were repeated once (Day 1 and 2; Day 6 and 7), respectively. Tumor volume was recorded within the treatment period. Obviously, Group 6 (Os/BC NPs + $hv$) exhibited the most obvious suppression effect on tumor growth as compared to other groups (Fig. 6g, h). Group 3 (Os NPs + $hv$) displayed slightly suppression effect on tumor volume. However, no statistical difference was found between Group 3 and the control (Groups 1) ($p = 0.1819$ by two-tailed Student's $t$-test). This result is consistent with the above finding of low cytotoxicity of Os NPs upon light irradiation, which can be explained by the limited phototoxicity of Os(bptpy)₂²⁺. Systemically administered free Cb (group 2) did not exhibit detectable anti-tumor effect due to its rapid hydolysis and short circulation time in the blood[40]. On Day 13, we euthanized the mice and excised tumors and organs for ex vivo characterizations. Tumor weight of Group 6 was significantly lower than other groups, demonstrating the excellent anti-tumor efficacy of Os/BC NPs with NIR light (Supplementary Fig. 31). Hematoxylin and eosin (H&E) staining assay was conducted to investigate the pathology of the tumors and organs. Obvious necrosis was found in the tumor tissues treated with Os/BC NPs + $hv$, while negligible cell apoptosis/tissue necrosis were observed in other groups (Fig. 6i).

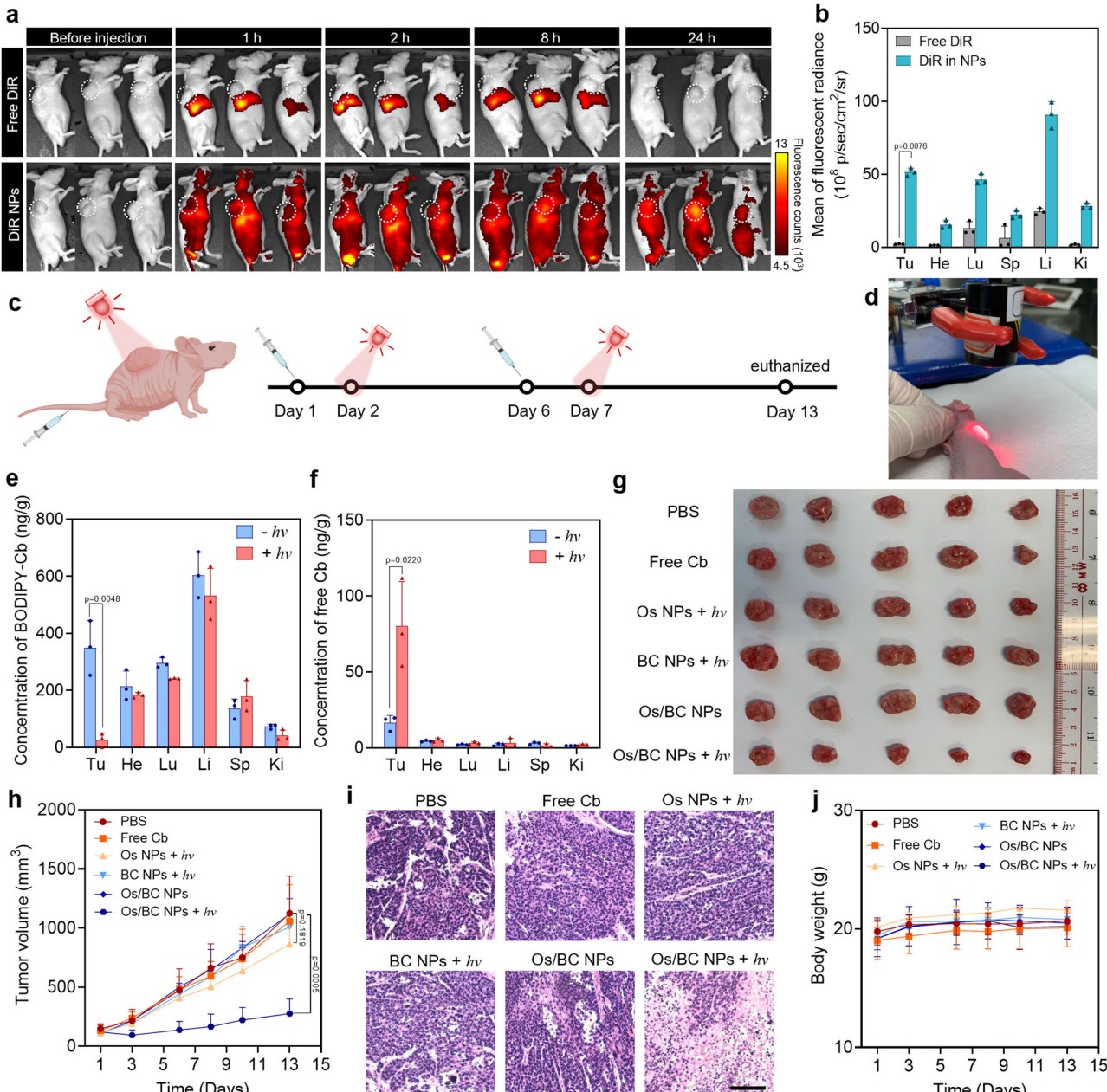

**Fig. 6 | Biodistribution and anti-tumor efficacy of the nanoparticles.**
**a** Representative IVIS fluorescence images of the mice after injection of free DiR and DiR NPs within 24 h. White dashed circles indicate tumor areas. **b** Quantitative analysis of biodistribution in major organs and tumors determined by IVIS. Tu, He, Lu, Sp, Li, and Ki represent tumor, heart, lung, spleen, liver, and kidney, respectively. Data are presented as mean ± SD, $n = 3$ mice per group. **c** Schematic illustration of the treatment schedule. Created with MedPeer (www.medpeer.cn). **d** Photograph of a mouse irradiated with NIR light at the tumor site. **e**, **f** Concentrations of BC and Cb in tumors and organs at 24 h after intravenous administration of Os/BC NPs with/without light irradiation on tumors. Data are presented as mean ± SD, $n = 3$ mice per group. **g** Photograph of tumors resected at Day 13 after different treatments. **h** Tumor volumes of each group. **i** Representative H&E staining of tumor sections of different treatment groups. Scale bar: 200 μm. Light irradiation: 690 nm, 300 mW/cm², 10 min. **j** Body weight of each group. In **h** and **j**, data are presented as mean ± SD, $n = 5$ mice per group. Statistical differences in **b**, **e**, **f**, **h** were analyzed by two-tailed Student's *t*-test. Source data are provided as a Source Data file.

Furthermore, no obvious tissue damage was observed in the major organs including the heart, lung, liver, spleen, and kidney in all treated mice (Supplementary Fig. 32). Also, no significant change in body weight was observed (Fig. 6j). Urea, creatinine, alanine aminotransferase (ALT) and aspartate aminotransferase (AST) in serum were determined and no systemic toxicity was observed except for the free Cb-treated group (Supplementary Fig. 33). Besides, as shown in Supplementary Fig. 34a, the free Cb treatment resulted in hemolysis, while no hemolysis (hemolysis ratio <5%) was observed after the nanoparticle treatment. The genotoxicity was evaluated by micronucleus assay and the bone marrow cell micronucleus number in the free Cb-treated mice was much higher than that of the other groups (Supplementary Fig. 34b, c). All these results indicated that the light-activatable prodrug nanoparticles exhibited less toxicity than the systemic administrated Cb.

## Maximum tolerated dose and renal & hepatic toxicity
Following the in vivo efficacy evaluation, the maximum tolerated dose (MTD) was determined. As shown in Supplementary Figure 35, mice exhibited tolerability for Os(bptpy)₂·2PF₆ under 24 mg/kg, but not

under 48 mg/kg. Notably, the Os NPs were well-tolerated in all the groups from 6 to 48 mg/kg (on the basis of Os complex), which showed neither body weight changes nor animal distress after systemic administration of Os NPs. The result indicates a high MTD of Os NPs that is over 48 mg/kg in mice. The renal and hepatic toxicity were determined by measuring the urea, creatinine, ALT, and AST in serum of the mice treated with free Os and Os NPs, separately (Supplementary Fig. 36). No significant renal and hepatic toxicity was observed.

## Discussion

In summary, we have developed a prodrug photolysis strategy based on the one-step energy transfer from STPS to BODIPY-based prodrugs, overcoming the limitation of required photon energy that must be higher than the $S_1$ state of PS. This process depends on the direct activation of STPS to the triplet excited state, followed by energy transfer from the $T_1$ state of STPS to that of prodrugs. Such a strategy allows the utilization of low-energy photons, such as NIR photons. Besides the prolonged excitation wavelength, this strategy demonstrated many other strengths, such as low light irradiance and high photolytic yield, which are presumably explained by the reduced energy loss and less photodamage during the photolysis process. Notably, such an upconversion-like photolysis strategy can be highly modular. A series of prodrugs were fabricated for NIR light-triggered optochemical control of a board range of bioactive molecules, including anti-cancer drugs, anti-inflammation drugs, anesthesia agents, and biogenic amines. The triplet state lifetime of STPS is important in the energy transfer process between STPS and prodrugs. It was reported that long-triplet-lifetime Os PS can be prepared by covalently conjugating moieties with large π-conjugation structure and long triplet lifetime, such as perylene, which enhanced the intramolecular triplet energy transfer between the metal centre and ligand by excited-state thermal equilibrium[41,42]. Other strategies, such as discovering STPSs with other metal centres[43], modulating chromophore environment[44], and stabilizing triplet states by non-covalent interactions[45], will properly be workable to promote the photophysical performance of the STPSs and, finally, increase the efficiency of the energy transfer-based photolysis reactions in the future.

Moreover, PLA-PEG micellar nanoparticles can enclose PS and prodrug, which allows NIR light-triggered drug release for effective cancer therapy. The strategy of this study that uses light to activate prodrugs enables the accurately controllable release of drugs with different pharmacological functions. Thus, the therapeutic effect of such prodrug systems can be diverse to adapt to different therapeutic requirements. Besides, such photolysis reaction in the nanoparticle system can work well in normoxia environments and the required excitation power density is relatively low (100 mW/cm²).

In all, we have verified a simplified mechanism for NIR light-triggered photolysis via one-step energy transfer and a nanoparticle-based method for biomedical applications. This study provides insights for developing photoactivatable systems for the application of photopharmacology and photoresponsive drug delivery. In the future, further developments of this strategy can be anticipated by developing PSs that exhibits longer excitation wavelengths (such as NIR-II light) with deep penetration, more efficient S-T transition capability, longer triplet state lifetime, and more satisfying biocompatibility.

## Methods
### Ethical statement
This study complies with all relevant ethical regulations. The animal experiment and procedures were approved by the Committee on the Use of Live Animals in Teaching & Research (CULATR), The University of Hong Kong (Protocol No. 4381-17).

## Materials
p-Nitrophenyl chloroformate were obtained from Sigma-Aldrich (Steinheim, Germany). Osmium (III) chloride hydrate, 4'-(4-bromophenyl)-2,2':6',2''-terpyridine, acetoxy-acetyl chloride, ammonium hexafluorophosphate, boron trifluoride etherate, 2,4-dimethylpyrrole pyridine, triethylamine, hydrochloric acid, N, N-diisopropylethylamine, 3-(4,5-dimethyl-2-thiazolyl)-2,5-diphenyl-2-H-tetrazolium bromide (MTT), and all the other chemicals were obtained from Dieckman (Shenzhen, China). Chlorambucil, dimethyl-xanthone acetic acid, naproxen, ibuprofen, indomethacin, 4-benzyloxycinnamic acid, tetracaine, dopamine, tyramine and homoveratrylamine were purchased from Bide Pharm (Shanghai, China). 2',7'-Dichlorofluorescin diace (DCFH-DA) and 4',6-diamidino-2-phenylindole (DAPI) were obtained from Thermo Fisher (Heysham, Lancashire, UK). Solvents, including dichloromethane, N, N-Dimethylformamide, dimethyl sulfoxide, acetonitrile, methanol, hexene, ethyl acetate, tetrahydrofuran, were obtained from Oriental Co., Ltd. (Hong Kong, China). Silica gel columns were purchased from Teledyne ISCO (Lincoln, USA). PLA$_{5k}$-mPEG$_{5k}$ was supplied by Ponsure Biological Co., Ltd. (Shanghai, China).

## Characterizations
CombiFlash Rf chromatography instrument was purchased from Teledyne ISCO (Lincoln, USA) for chemical purification. Light sources used in this study were purchased from Yuanming Laser (Ningbo, China) and Mightex (CA, USA). The purity verification and photolysis study were conducted with a high-performance liquid chromatography (HPLC) system (Agilent technologies, 1260), attached with C-18 columns which were obtained from Agilent (Santa Clara, California, USA). Nuclear magnetic resonance (NMR) spectroscopy was collected by Bruker AVANCE III HD 500 spectrometer (Bremen, Germany). MALDI-TOF MS spectrum was collected by Bruker ultrafleXtreme system (Bremen, Germany). Fluorescence, phosphorescence, and UV-vis spectra were collected by a SpectraMax M4 microplate reader (Molecular Devices, CA, USA). The particle size and surface charge were recorded by a dynamic light scattering (DLS) device, Zetasizer Nano-ZS90 (Malvern instruments, UK). Transmission electron microscope (TEM) imaging was conducted on a Hitachi HT7700 TEM (Tokyo, Japan). ACEA NovoCyte Quanteon flow cytometer (ACEA Biosciences, CA, USA) was used for flow cytometry. In vivo and ex vivo imaging of the mice were conducted with an In Vivo Imaging System (PerkinElmer, USA). Devices for animal study including balance and vernier caliper were kindly provided by The Centre for Comparative Medicine Research (CCMR), The University of Hong Kong.

## DFT and TD-DFT calculations
Briefly, density functional theory (DFT) and time-dependent DFT (TD-DFT) were used with a B3LYP functional for geometry optimizations and energy computation, respectively. The LANL2DZ basis set was used for the iodine and osmium atoms, while the 6-31 G(d) basis set was used for the other atoms. After optimizing the geometry of the ground state, excited singlet state, and excited triplet state of BODIPY photocage, Os and BC, the energy level were calculated and recorded. The vertical excitation energies of the singlet states and triplet states were computed at the optimized ground state geometries. All DFT and TD-DFT calculations were carried out in the Gaussian 16 C.01 package.

## Measurement of energy transfer quenching rate constants ($k_{TTET}$)
The energy transfer quenching rate constants ($k_{TTET}$) were measured by Stern-Volmer experiment. As the representative of BODIPY prodrugs, BODIPY-Cb (compound 4) was used as the energy receptor while Os(bptpy)$_2$²⁺ was the energy donor during the quenching process. 10 μM of Os(bptpy)$_2$²⁺ solutions were prepared in the present of different concentrations of BODIPY-Cb (0, 1, 2, 3, 5, 7.5 μM), of which the solvent was mixed by 88% methanol, 2% acetone and 10%

dichloromethane. Oxygen was removed by nitrogen bubbling (50 mL/min) for 10 min.

Quenching constants ($k_q$) of Os(bptpy)$_2^{2+}$ in the presence of BODIPY-Cb were calculated by the following Eq. (1):

$$\frac{I_0}{I} = 1 + k_q[Q] \tag{1}$$

($I_O$: phosphorescence intensity of the Os(bptpy)$_2^{2+}$ solution; $I$: phosphorescence intensity of Os(bptpy)$_2^{2+}$ solution in presence of BODIPY-Cb; [Q]: concentration of BODIPY-Cb.)

Energy transfer quenching rate constants of this TTET process, quantified as $k_{TTET}$, can be calculated based on the below Eq. (2):

$$k_{TTET} = \frac{k_q}{\tau_0} \tag{2}$$

($\tau_O$: phosphorescence lifetime of Os(bptpy)$_2^{2+}$ in N$_2$-saturated solution without quencher.)

For Os(bptpy)$_2^{2+}$, $\tau_0 = 0.20\ \mu s$[23]. The $k_q$ of Os(bptpy)$_2^{2+}$ in present of BODIPY-Cb was determined to be $(5.491 \pm 0.282) \times 10^4\ M^{-1}$. Thus, the energy transfer quenching rate constant of the TTET between Os(bptpy)$_2^{2+}$ and BODIPY-Cb was calculated as $(2.718 \pm 0.140) \times 10^{11}\ M^{-1}\ s^{-1}$.

## Quantitative analysis of BODIPY prodrug photolysis

The photolysis efficiency of BODIPY prodrugs were compared in different conditions, including different excitation wavelengths (690 nm or 530 nm), different ratios of Os(bptpy)$_2^{2+}$, and different oxygen contents. A 100 μL mixture of BODIPY-Cb (compound 4) (100 μM) and Os(bptpy)$_2^{2+}$ (0 eq. (0 μM), 0.05 eq. (5 μM), 0.1 eq. (10 μM)) was saturated with nitrogen or not and irradiated by 690 nm light (100 mW/cm$^2$, 5 min) or 530 nm light. The solvent was 88% methanol mixed by 10% dichloromethane and 2% acetone. For the N$_2$-saturated groups, after dissolving the molecules, the solution was degassed with a vacuum pump for 10 min followed by gentle N$_2$ blowing. After light irradiation, the resulted mixtures were refilled to 100 μL with methanol to avoid the error caused by solvent evaporation. The resulted solution was loaded onto the sampler and analyzed by high-performance liquid chromatography. The attached elution method (Supplementary Table 2) was used to separate free drug, BODIPY photocage and prodrug on the C18 column.

## Measurement of the photoreaction quantum yields

Quantum yields of the photoreactions (Φ) are defined as:

$$\phi = \frac{\text{number of reacted molecules per time unit}}{\text{number of absorbed photons per time unit}} \tag{3}$$

The number of reacted molecules (e.g., consumption of prodrugs or generation of free drugs) were determined by HPLC. The absorbed photon number was determined by Reinecke's salt actinometry at excitation wavelengths of 530 nm and 690 nm[46,47]. The power densities were 50 mW/cm$^2$ (530 nm) and 100 mW/cm$^2$ (690 nm).

## Fabrication and characterization of nanoparticles

Flash nanoprecipitation method was used to fabricate the photosensitizer/prodrug-loaded nanoparticles. The stock solutions of PLA$_{5k}$-mPEG$_{5k}$ (200 mg/mL in acetone), BODIPY-Cb ($3 \times 10^{-2}$ M, 24.2 mg/mL in DMSO) and Os(bptpy)$_2 \cdot$PF$_6$ ($10^{-3}$ M, 1.26 mg/mL in DMSO) were prepared. Different formulations (Os NPs, BC NPs and Os/BC NPs) were then prepared with optimized ratios of components. Briefly, 224 μL of the mixed stock solution was added into 1.8 mL N$_2$-saturated water under vortexing. The resulted solutions were added to dialysis bags (M$_W$: 3400 Da) and dialyzed against 4 L of water for 24 h. The water out

of the dialysis bag was renewed every 8 h. The resulted solution of NPs was collected and filtrated with 220 nm filter. Finally, the NPs solutions were concentrated by ultrafiltration (Mw = 100 kDa, 2000 × g, 10 min) and stored at 4 °C until use. The concentrations of each component were measured by using HPLC. Loading capacity and encapsulation efficiency were calculated as follows:

$$\text{Loading capacity (\%)} = \frac{\text{weight of loaded payload}}{\text{weight of nanoparticles}} \times 100\% \tag{4}$$

$$\text{Encapsulation efficiency (\%)} = \frac{\text{weight of loaded payload}}{\text{weight of fed payload}} \times 100\% \tag{5}$$

## Measurement of prodrug photolysis in nanoparticles

Aqueous solution of Os NPs, BC NPs and Os/BC NPs were diluted to $10^{-3}$ M (on basis of BODIPY-Cb) and added to 1.5 mL tubes. 690 nm light (100 mW/cm$^2$) was applied topically onto the solution at room temperature for 0-30 min. At each time point, 100 μL sample was collected and dispersed with equal volume (100 μL) of acetonitrile. Then HPLC was used to analyze the prodrug consumption and drug release yield.

## Cell culture

Human Cervical Adenocarcinoma cells (HeLa) were purchased from the Cell Bank of Chinese Academy of Sciences (China). Cells were cultured in DMEM (Gibco) supplemented with 10% FBS (Gibco) and 100 units/mL antibiotics (Penicillin-Streptomycin, Gibco) at 37 °C in a 5% CO$_2$ humidified atmosphere.

## Cytotoxicity analysis

Cell viabilities were determined by MTT assay. Hela cells were cultured on 96-well plates at a primary density of 5000 cells/well in 100 μL complete DMEM medium and incubated for 24 h. The medium was replaced with the formulations-contained medium (Os NPs, BC NPs and Os/BC NPs) at different concentrations (0–50 μM on basis of BODIPY-Cb). After 4-h incubation, the cells were irradiated by NIR light (690 nm, 100 mW/cm$^2$) for 30 min. MTT solution (10 μL/well) was added after 24 h of incubation. After 3 h, the medium was discarded, and DMSO (100 μL) was added into each well. OD490 values were recorded by plate reader for the calculation of cell viability.

## Nitric Oxide Content detection in RAW 264.7 cells

The concentration of nitric oxide of LPS-stimulated RAW 264.7 cells was determined by Griess reagent. RAW 264.7 cells in complete DMEM medium were treated with LPS (20 ng/mL) and the formulations (PBS, Os/BI NPs, Os/BN NPs) at different concentrations (1–5 μM on basis of prodrugs). For light-triggered groups, NIR light (690 nm, 100 mW/cm$^2$, 30 min) was applied after 4 h-incubation. After 24 h, 100 μL culture supernatant was collected and reacted with an equal volume of Griess reagent (40 mg/mL in H$_2$O). The absorbance was measured at 540 nm to determine the NO concentrations.

## Live/dead cell staining

HeLa cells were seeded in confocal dishes at a density of 10000 cells/well and treated with PBS, Os NPs, BC NPs, Os/BC NPs with an equivalent concentration of BODIPY-Cb at 10 μM. For light irradiation, cells were irradiated after 4 h by 690 nm light (100 mW/cm$^2$) for 30 min, followed by 24 h incubation. After washing with PBS and replacing the medium, Calcein-AM and PI were added and the cells were observed by confocal laser scanning microscopy ($\lambda_{ex} = 488$ nm, 560 nm).

## Cell apoptosis analysis

HeLa cells were seeded in 6-well plates at a density of 50000 cells/well and treated with free Cb or IR783/BC NPs PBS, Os NPs, BC NPs, Os/BC

NPs with an equivalent concentration of BODIPY-Cb at 10 µM for 4 h. Then the cells in the irradiation group were irradiated by light (690 nm, 100 mW/cm$^2$) for 30 min. After 24-h incubation, cells were washed with PBS for 3 times, collected by trypsin digestion, and stained with Annexin-V/FITC apoptosis kit. The cell suspension solutions were analyzed by flow cytometer ($\lambda_{ex}$ = 488 nm, 560 nm). A figure exemplifying the gating strategy for flow cytometry experiments is provided as Supplementary Fig. 51.

### Animals

BALB/c nude mice (age 4 weeks, about 20 g) were used for tumor implantation and further study. All mice were obtained from the Experimental Animal Center of University with access to food and water ad libitum and maintained under pathogen-free condition. Other environmental conditions were: photo-period control with 12-h light/ 12-h dark cycle; temperatures of 16–26 °C with 30–70% humidity; 100% fresh air supply with 15 air changes per hour.

HeLa tumor-bearing mice were obtained by subcutaneously injecting HeLa cells for tumor implantation. HeLa cancer cells were collected by cell culture and trypsin digestion. The cells were washed by PBS and resuspended in DMEM medium without serum, in which collagen and Matrigel were added. A total of $2 \times 10^6$ cells were dispersed in 100 µL mixture, which was subcutaneously injected into the mice at underarm area. After 7 days incubation, the mice with tumor were randomly divided into groups for in vivo experiments. The maximal tumor size of 1500 mm$^3$ was permitted by CULATR, The University of Hong Kong.

### In vivo biodistribution

The biodistribution of free drug or nanoparticles in the HeLa tumor-bearing mice was measured by an in vivo fluorescence imaging system. DiR (1,1-dioctadecyl-3,3,3,3-tetramethylindotricarbocyanine iodide) was used as the fluorescent dye for labeling the PLA-mPEG NPs during living imaging. 6 mice with HeLa tumors (volumes at about 200 mm$^3$) were randomly divided into two groups and treated with free DiR or DiR in PLA-mPEG NPs via intravenous injection with a dose of DiR at 100 µg/kg. The in vivo fluorescence imaging was performed at 1, 2, 8, and 24 h post injection and anesthesia at each time point. At 24 h, all the mice were euthanized. Tumors and major organs (heart, liver, spleen, lung, kidney) were excised for ex vivo imaging ($\lambda_{ex}$ = 710 nm, $\lambda_{em}$ = 780 nm).

### In vivo light-triggered prodrug activation and tumor inhibition

The anti-tumor efficacy of the formulations in the presence or absence of light was investigated with HeLa tumor-bearing mice. 7 days after tumor implantation, the mice with tumors at the volumes at around 100 mm$^3$ were randomly divided into 6 groups (5 mice per group). Different treatments were administrated to each group: (1) PBS; (2) free Cb; (3) Os NPs plus light irradiation; (4) BC NPs plus light irradiation; (5) Os/BC NPs; (6) Os/BC NPs plus light irradiation. The formulations were injected intravenously on Day 1 and Day 6, of which the dose was set as 8 mg/kg. For the groups with light irradiation, light irradiation (690 nm, 300 mW/cm$^2$, 10 min) were performed 24 h post injection (on Day 2 and Day 7). Tumor sizes and body weights were measured during the period, and the tumor volume was calculated as V = 1/2 × width$^2$ × length. On Day 13, all the mice were euthanized, and the tumors and major organs were excised and sliced for H&E staining and histochemical analysis. All the histological study were kindly performed in blinded fashion by a pathologist from the Department of Pathology at the University of Hong Kong.

LC/MS/MS was used to quantify the prodrug activation in tumor. Briefly, the Os/BC NPs were injected into the tail vein of tumor-bearing mice (tumor volume at around 200 mm$^3$, $n$ = 3 independent tests), of which the dosage was the same as the therapeutic settings. For the light irradiation group, NIR light (690 nm, 300 mW/cm$^2$, 10 min) were applied on the tumor area 24 h post-injection. Then the mice were euthanized. Tumors and major organs were collected, weighted, and homogenized. The prodrug (BC) and free drug (Cb) were extracted by acetonitrile from the tissue homogenates, followed by the centrifugation at 10400 × g for 10 min, and the supernatant was collected and filtered through 220 nm filter before injecting into LC/MS/MS system (Agilent 1290/AB SCIEX 3200 QTRAP). The concentrations of free Cb and BC in tissues were determined by multiple reaction monitoring (MRM) of ions.

### Hemolytic test

The hemolytic test was conducted by adding free Cb or Os/BC NPs at different concentrations (1-10 µM, on basis of Cb, diluted in PBS) into red blood cells (RBC) suspension, followed by 2-h incubation at 37 °C. PBS, the negative control, and distilled water, the positive control, were added meanwhile. Mixtures were then centrifuged at 3000 g for 10 min. The hemolysis was determined by the absorbance at 542 nm.

### Micronucleus assay

Micronucleus assay was conducted based on the reported method[48]. After the treatments with different formulations, bone marrow cells from femurs of the mice were collected and sectioned. After fixation by methanol for 10 min, the cells were stained with 10% Giemsa solution for 30 min and polychromatic erythrocytes are scored for the incidence of micronuclei (per 500 cells of each mouse).

### Maximum tolerated dose and renal & hepatic toxicity

The maximum tolerated dose (MTD) of free Os and Os NPs was measured[49]. Briefly, MTD was determined on BALB/c mice (6-8 weeks) by intravenously administering a series of drug doses (6 mg/kg, 12 mg/ kg, 24 mg/kg, 48 mg/kg; on basis of Os). For free Os, the solution for dispersing the photosensitizer was 4% DMSO + 0.05% Tween 80 in saline. The mice were observed constantly for 2 hours, then periodically for up to two weeks. The animal distress was recorded, and the body weight was measured every two days.

After 2 weeks post-injection, the mice were euthanized, and the serum was collected. Blood urea, creatinine, alanine aminotransferase (ALT) and aspartate aminotransferase (AST) were measured to evaluate the renal and hepatic toxicity.

### Reporting summary

Further information on research design is available in the Nature Portfolio Reporting Summary linked to this article.

## Data availability

Data generated or analyzed in this study are all included in the article and Supplementary Information file. The full image dataset is available from the corresponding author upon request. Source data are provided with this paper.

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

## Acknowledgements

This work was supported by the National Natural Science Foundation of China, Excellent Young Scientists Fund (No. 82222903) to W.W., National Natural Science Foundation of China (No. 22103039) to W.L., Health and Medical Research Fund of Hong Kong (No. 07181936) to W.W., Research Grants Council of Hong Kong (Early Career Scheme, No. 27115220) to W.W., Li Ka Shing Faculty of Medicine (Start-up Fund) to W.W., Ming Wai Lau Centre for Reparative Medicine Associate Member Program to W.W., Natural Science Foundation of Jiangsu Province (No. BK20210583) to W.L., Natural Science Foundation of Jiangsu Higher Education Institutions of China (No. 21KJB150013) to W.L. We acknowledge the assistance of Faculty Core Facility, Li Ka Shing Faculty of Medicine, The University of Hong Kong. We acknowledge Dr. Minling Zhong in The Chinese University of Hong Kong for her assistance in theoretical calculations.

## Author contributions

K.L., W.L., and W.W. designed the project. K.L., W.L., and Z.W. performed the chemical synthesis and photolysis experiments. K.L., K.C., Y.Z., and Z.W. conducted cell experiments and animal experiments. N.F. performed LC/MS/MS experiments. K.L., W.L., and F.L. finished the theoretical calculations. W.W. supervised the study. K.L. and W.L. wrote the manuscript and all the authors reviewed and approved the manuscript.

## Competing interests

W.W. and K.L. are inventors on US provisional patent application 63/506,991, submitted by The University of Hong Kong, which covers the preparation method of prodrugs and the nanoparticles in this study and the in vivo applications. All other authors declare no competing interests.
