## [Peer Review File · Nature Communications]

Reviewers' Comments:

Reviewer #1:

Remarks to the Author:

The authors introduce a NIR light-driven photocleavage system that utilizes an Os complex and a Bodipy-based prodrug via triplet sensitization, with demonstrated *in vitro* and *in vivo* applications. However, the reviewer contends that the main contribution is not significant compared to existing methods, which already employ metalloporphyrin and Bodipy-based prodrugs activated by long wavelength light. Specifically, the emphasis on a single step of energy transfer is not seen as an important advancement, as the ISC efficiency of existing photosensitizers is typically already close to 100%. Furthermore, the Os complex is rather toxic *in vivo* and not ideal for biological applications, featuring weak absorption and S-T transition in the NIR region, along with a short triplet state lifetime. The applications presented are also not novel compared to previously reported methods. As a result, the reviewer does not recommend the paper for acceptance in *Nature Communications*.

Reviewer #2:

Remarks to the Author:

The authors report the activation of prodrugs by irradiation of near-infrared light and their therapeutic effects on cancer using a mouse model. They used a triplet sensitizer that exhibits S-T absorption as previously reported and succeeded in activating the prodrug by irradiation with relatively weak near-infrared light. In addition, they successfully released the anticancer drug at the cellular and mouse levels by encapsulating the dye and prodrug in micelles. The near-infrared light-absorbing triplet sensitizers, encapsulation of dyes into micelles, and release of various anticancer drugs have all been reported, so it is difficult to say that this is a novelty. The methodology for the release of anticancer drugs by near-infrared light is useful though. The following individual points need to be addressed.

1. The use of the term "upconversion-like" from the title to the text is inappropriate and misleading and should be reworded. Upconversion is a term used to describe the generation of an energy state higher than the excitation energy, while the present system is all downconversion and there is no element of upconversion.
2. The present system is so-called triplet sensitization, and the target of comparison is not upconversion but photolysis by triplet sensitization. They should explain the superiority of this system over the many other photolysis systems that have been reported in the past.
3. It is reported that near-infrared photolysis is more efficient than green-light and red-light photolysis, but the reason for this is not fully understood.
4. Around line 220, it is explained that photolysis does not occur due to low ISC efficiency and short T1 lifetime without iodine atoms, but this explanation is unreasonable. The ISC efficiency should not be relevant for the photolysis efficiency, since the proposed mechanism does not require ISC for the BODIPY derivative. In addition, the T1 lifetime is likely to be rather shortened by the heavy atom effect when the iodine atom is modified. This should be evaluated by transient absorption measurements.

Reviewer #3:

Remarks to the Author:

Near-infrared (NIR) light-triggered drug release is preferable for drug delivery and precise disease treatment due to its deep tissue penetration and low phototoxicity. This manuscript by Wang et al developed a new one-step triplet-triplet energy transfer strategy for NIR light-triggered prodrug photolysis, based on Os (II) bromophenyl terpyridine complex as a photosensitizer and a wide range of boron-dipyrromethene (BODIPY)-based prodrugs and bioactive molecules. Moreover, a micellar nanosystem encapsulating both photosensitizer and prodrug has been demonstrated the practicality for *in vivo* cancer therapy in normoxia aqueous environment. The present work would appeal to a broad readership of the journal and I support publication after minor revision.

1. Please provide DFT and TD-DFT calculations to verify the assignment of the S1 and T1 states of

Os (II) bromophenyl terpyridine complex in the UV-vis absorption spectrum.

2. Please explain how to determine the amount of released Chlorambucil drug in vivo.

Response to the comments

Reviewer #1:

The authors introduce a NIR light-driven photocleavage system that utilizes an Os complex and a Bodipy-based prodrug via triplet sensitization, with demonstrated in vitro and in vivo applications. However, the reviewer contends that the main contribution is not significant compared to existing methods, which already employ metalloporphyrin and Bodipy-based prodrugs activated by long wavelength light. Specifically, the emphasis on a single step of energy transfer is not seen as an important advancement, as the ISC efficiency of existing photosensitizers is typically already close to 100%. Furthermore, the Os complex is rather toxic in vivo and not ideal for biological applications, featuring weak absorption and S-T transition in the NIR region, along with a short triplet state lifetime. The applications presented are also not novel compared to previously reported methods. As a result, the reviewer does not recommend the paper for acceptance in *Nature Communications*.

Response: We acknowledge the reviewer to point out concerns of our study, including the high ISC efficiency of the existing photosensitizers, safety concern of Os complex, and limited photophysical properties. It is very important for us to consider these points and address the concerns to improve our study. Here we provide point-by-point response to the comments as below:

“However, the reviewer contends that the main contribution is not significant compared to existing methods, which already employ metalloporphyrin and Bodipy-based prodrugs activated by long wavelength light.”

Response: We acknowledge the reviewer for the comment. For the photosensitizers employed in the reported TTA upconversion and upconversion-like process (ref. 20 in the manuscript), it is necessary to populate S_1 state before generating T_1 state through ISC process. Thus, the required photon energy to activate such a reaction should be higher than the T_1 energy level of the photosensitizer. Nevertheless, in this study, the required photon energy decreased to the T_1 energy level of the photosensitizer, since the Os complex could be directly activated to the T_1 state according to the S-T absorption process. As a result, we can greatly prolong the wavelength of excitation light even to 750 nm, which could hardly be achieved by utilizing the traditional

photosensitizer of metalloporphyrins. And such NIR light exhibits deeper tissue penetration and lower phototoxicity in biological systems for prodrug activation.

Figure 1. Schematic illustration of the mechanisms of the reported photolysis strategies and the NIR light-triggered upconversion-like photolysis with one-step energy transfer in this work. The red arrows represent the required energy for initiating the photolysis process (required lower photon energy) of each route.

“The emphasis on a single step of energy transfer is not seen as an important advancement, as the ISC efficiency of existing photosensitizers is typically already close to 100%.”

Response: We agree that the intersystem crossing (ISC) efficiency of some photosensitizers is high since most of the well-designed transition metal complexes can generate high population of triplet states (Coordination Chemistry Reviews 417 (2020): 213371). Nevertheless, the advancement of our strategy is not only bypassing the ISC process to eliminate energy loss in ISC, but also reducing excitation energy from $S_0 \rightarrow S_1$ to $S_0 \rightarrow T_1$ to activate photolysis reaction with 650 nm~750 nm NIR light (**Figure 1**, modified from Figure 1 in the revised manuscript). Moreover, we found that utilizing low-energy NIR photons and simplifying energy transfer processes can reduce photodamage of the prodrugs and unexpected relaxation of the excited states. Therefore, the demonstrated photolysis reaction of BODIPY prodrugs can achieve a higher yield of free drug with NIR light than that triggered by green light or red light.

“Furthermore, the Os complex is rather toxic in vivo and not ideal for biological applications, featuring weak absorption and S-T transition in the NIR region, along with a short triplet state lifetime. The applications presented are also not novel compared to previously reported methods.”

Response: We acknowledge the reviewer to point out the concerns of Os complex in biological applications. For evaluating the biosafety, we have analyzed the H&E staining slices of the main organs of the post-treatment mice (**Figure 6G**), which showed no obvious tissue damage in organs including heart, lung, liver, spleen and kidney. Furthermore, the body weight of mice had negligible change during the treatment (Figure S16), indicating satisfying safety of the formulations (Os NPs or Os/BC NPs) at the set dosage (about 6 mg/kg, on basis of Os complex) in tumor-bearing mouse model. The descriptions of such results can be found on page 20 as “*No obvious tissue damage was observed in major organs including heart, lung, liver, spleen, and kidney in all treated mice, indicating negligible in vivo toxicity of the treatments (Figure S22). Moreover, no significant change of the body weight was observed, further indicating low systemic side effects of our system (Figure 6G).*”.

Figure 6G. Body weight of the mice in different groups.

Figure. S16. H&E staining of major organ sections of the mice after different treatments.

Scale bar: 200 μ m.

To further address the safety concerns on Os complex, we did additional experiments and investigated the *in vivo* acute toxicity by measuring the maximum tolerated dose (MTD) to determine the safe amount of Os complexes while applying *in vivo*. Os(btpy)₂·2PF₆ (abbreviated as Os) and PLA-PEG nanoparticles encapsulating Os (abbreviated as Os NPs) were intravenously injected into BALB/c mice with different dosages (6, 12, 24, 48 mg/kg, on basis of Os complex), separately. The results showed that the Os NPs were well-tolerated in all the groups, indicating an MTD over 48 mg/kg (**Figure S25**). The free Os was tolerated at 24 mg/kg, while weight loss within 2 days was observed at the 48 mg/kg dose, indicating an MTD of 24 mg/kg. We further measured the blood urea and serum creatinine for renal function; alanine aminotransferase (ALT) and aspartate aminotransferase (AST) for hepatic function of the mice treated with free Os and Os NPs. No significant renal and hepatic toxicity was observed (**Figure S26**).

We added the figures (Figure S25-26) in supporting information and added one section at page 22 as below:

“Maximum tolerated dose and renal & hepatic toxicity

Following the in vivo efficacy evaluation, the maximum tolerated dose (MTD) was determined. As shown in Figure S25, mice exhibited tolerability for Os(btpy)₂·2PF₆ under 24 mg/kg, but not under 48 mg/kg. Notably, the Os NPs were well-tolerated in all the groups from 6 to 48 mg/kg (on basis of Os complex), which showed neither body weight change nor animal distress after systemic administration of Os NPs. The result indicates a high MTD of Os NPs that is over 48 mg/kg in mice. The renal and hepatic toxicity were determined by measuring the urea, creatinine, alanine aminotransferase (ALT) and aspartate aminotransferase (AST) in serum of the mice treated with free Os and Os NPs, separately (Figure S26). As a result, no significant renal and hepatic toxicity was observed.”

Moreover, the methodology of the additional experiments was added in the revised supporting information (Page 18).

Figure. S25. The body weight of the mice treated with saline with/without DMSO and Tween 80 (A), free Os (B) and Os NPs (C) from 6 to 48 mg/kg (on basis of Os complex) by i.v. injection within 14 days (# euthanized due to weight loss).

Figure. S26. Serum biochemistry data including (A) aminotransferase (ALT); (B) aspartate aminotransferase (AST); (C) urea, and (D) creatinine (CREA). The mice were treated with free Os or Os NPs from 6 to 48 mg/kg by i.v. injection. (# euthanized due to weight loss).

In addition, there have been many Os(II) complexes studied for biomedical applications (Chemical Society Reviews 47.3 (2018): 909-928; Chem. Commun., 2022, 58, 4825-4836). TLD1824 and TLD1829 were Os (II)-polypyridyl complexes with excellent PDT effects, which also exhibited relatively high MTDs at 6.25 mg/kg (TLD1824) and 47.0 mg/kg (TLD1829) after i.p. injection (Photochemistry and Photobiology 93.5 (2017): 1248-1258). Another study revealed that Os-4T (Os(phen)₂IP-4T, T = thiophene) exhibited a high MTD over 200 mg/kg, which is an excellent candidate for *in vivo* applications. Notably, analogue of Os-4T, TLD1433, showed excellent potential in clinical Ib trial (Chemical reviews 119.2 (2018): 797-828.). In our testing, Os(btpy)₂·2PF₆ and its nanoparticles (Os NPs) showed

comparable MTD (24 mg/kg for Os and over 48mg/kg for Os NPs). And, importantly, smaller amount of Os was used in our study, of which the dose of Os complex was about 6 mg/kg in Os NPs. Thus, neither side effects nor acute toxicity were observed. In the near future, we will also try to develop photosensitizers with enhanced singlet-triplet absorption and minimum side effects, and the toxicity of such metalloorganic complexes will be carefully controlled by rational ligand design, careful dosage managements and sufficient preclinical examinations.

The reviewer mentioned that S-T transition of Os complexes is weak. It can be explained by the fact that the direct S_0-T_1 absorption is theoretically spin-forbidden (Journal of the American Chemical Society 138.28 (2016): 8702-8705). Currently, only a few kinds of photosensitizers have been reported to exhibit such S-T transition (*Dyes and Pigments* 199 (2022): 110049). As the first proof-of-concept study of photolysis reaction based on S-T transition, the photolytic efficiency in this study was satisfying by utilizing the S-T absorption of $Os(btpy)_2 \cdot 2PF_6$, which demonstrates the superiority of our one-step energy transfer strategy. Further development can be seen by developing novel metal-organic complexes with strong S-T absorption and long triplet lifetime, which would increase the efficiency of such photolysis systems.

Moreover, the reviewer pointed out that the application of our systems is not novel. It should be noted that this study aims to verify the feasibility of using one-step energy transfer to trigger prodrug photolysis. The efficacy was evaluated on a conventional tumor model since it was the most common yet solid model to verify photoactivation of prodrugs. Besides, we also developed the modular design of this system, of which the released payloads can be different bioactive molecules, which may hopefully be utilized to achieve photocontrol of many other biological processes.

In all, we sincerely appreciate the reviewer pointing out the concerns. For biocompatibility, we did additional testing, of which the results indicate satisfying compatibility of our formulations. For the photophysical properties of Os, we agree that developing photosensitizers with strong S-T absorption and longer triplet lifetime can further increase the efficiency of this system. To emphasize this, we added one sentence at the end of the discussion section (Page 24): “*In the*

future, further development and clinical translation of this strategy can be seen by developing photosensitizers that exhibits efficient S-T transition capability, long triplet state lifetime, and satisfying biocompatibility.”

Reviewer #2:

The authors report the activation of prodrugs by irradiation of near-infrared light and their therapeutic effects on cancer using a mouse model. They used a triplet sensitizer that exhibits S-T absorption as previously reported and succeeded in activating the prodrug by irradiation with relatively weak near-infrared light. In addition, they successfully released the anticancer drug at the cellular and mouse levels by encapsulating the dye and prodrug in micelles. The near-infrared light absorbing triplet sensitizers, encapsulation of dyes into micelles, and release of various anticancer drugs have all been reported, so it is difficult to say that this is a novelty. The methodology for the release of anticancer drugs by near-infrared light is useful though.

Response: We acknowledge the for the comment. As for the novelty, this new photolysis strategy exhibits both prolonged excitation wavelength (reach the NIR window to achieve green light-responsive photolysis) and simpler energy transfer process (only one step), which can't be achieved by the previous photolysis mechanisms. Unexpectedly, the demonstrated photolysis reaction of BODIPY prodrugs can achieve a higher yield of free drug with NIR light than that triggered by green light or red light. The detailed responses to the reviewer's comments are as below:

1. The use of the term “upconversion-like” from the title to the text is inappropriate and misleading and should be reworded. Upconversion is a term used to describe the generation of an energy state higher than the excitation energy, while the present system is all down conversion and there is no element of upconversion.

Response: We appreciate the reviewer's suggestion. In our previous point of view, “upconversion-like” is not the real “upconversion” since we have employed long-wavelength light, such as NIR light, to activate short-wavelength light-responsive molecules, which was just like the outcomes of upconversion systems. However, no upconverting emission was generated during this process. To eliminate the misleading, we revised the title of the manuscript

as “*Near-infrared Light-triggered Prodrug Photolysis by One-step Energy Transfer*” in the revised version.

2. The present system is so-called triplet sensitization, and the target of comparison is not upconversion but photolysis by triplet sensitization. They should explain the superiority of this system over the many other photolysis systems that have been reported in the past.

Response: We appreciate the reviewer’s comment. We agree that our study is the photolysis based on the triplet sensitization process. The superiorities of this system over traditional systems can be summarized as: 1). Long-wavelength light triggerable. The wavelength of excitation light can be prolonged to near infrared region (650 nm~750 nm) by the novel energy transfer mechanism reported in this study. 2). High photolytic yield. The application of low-energy photons of long-wavelength light reduced the photobleaching of prodrugs (*refer to our response to comment 3*), thus the photolytic yields of free drugs were increased.

In addition, we agree that it is necessary to compare this study with other photolysis systems. Some long-wavelength light-activatable photolysis strategies are listed below. 1). Increasing the absorption wavelength through molecular modifications. Usually, the absorption wavelength can be increased by expanding its π conjugation or substitution of chemical groups (Chem. Soc. Rev. 2015; 44: 3358). However, it was time and labor consuming, while the chemical modifications on photocages may also affect their photolysis efficiency (Chem. Rev. 2013, 113: 119). 2). Two-photon excitation. Some photocages can absorb two coherent long-wavelength photons for photolysis reaction. However, the two-photon excitation of photocages requires femtosecond pump laser with high irradiance (10^6 W/cm² or above) and the reactions only occur at the laser focal point (Journal of Photochemistry and Photobiology C: Photochemistry Reviews, 48 (2021): 100423), which limits the biomedical applications. 3). Utilizing upconversion systems. Upconversion systems, such as lanthanide-doped upconversion nanoparticles (UCNPs) and triplet-triplet annihilation upconversion (TTA-UC) systems, can convert long-wavelength light into short-wavelength light. The upconversion systems undergo multi-step and multi-photon energy transfer, of which the photolysis efficiency highly depends on the internal energy transfer efficiency. Compared to these photolysis strategies, the one-step energy transfer mechanism in this study does not need the chemical

modifications on the photocage. Moreover, less energy transfer steps (specifically, only one step) are needed in our system as compared to those upconversion systems, which can reduce the energy loss and photobleaching of prodrugs and increase the photolytic efficiency. In the revised manuscript, the comparison between this strategy and other photolysis strategies can be seen in the second paragraph of the *Introduction Section* (page 3):

*“There have been strategies to achieve long-wavelength light-triggered prodrug photolysis, including 1). Increasing the absorption wavelength through molecular modifications; 2). Two-photon excitation; 3). Photon upconversion systems. Usually, the absorption wavelength can be increased by expanding its π conjugation or substitution of chemical groups. However, it was time and labor consuming, while the chemical modifications on photocages may also affect their photolysis efficiency.^{13,14} Moreover, the two-photon excitation of photocages requires femtosecond pump laser with high irradiance (10^6 W/cm² or above) and the reaction can only occur at the laser focal point.¹⁵ Lanthanide-doped upconversion nanoparticles (UCNPs) have emerged as reliable platforms for turning NIR light into UV/visible light, thus enabling long-wavelength light to activate short-wavelength light-responsive PPGs.^{16,17} However, the required excitation power density is still relatively high (10^1 - 10^4 W/cm²), since UCNPs exhibit low absorption coefficient and cross-sections, and the efficiency of luminescence resonance energy transfer (LRET) between UCNP and PPGs remains unsatisfactory. Triplet-triplet annihilation-based upconversion (TTA-UC) is another strategy for long-wavelength light-triggered photolysis, which depends on multi-step energy transfer between photosensitizer and annihilator to produce upconverted photons (**Figure 1A**).^{18,19} TTA-UC enabled the utilization of low-irradiance long-wavelength light (10^{-3} - 10^{-1} W/cm²), however, the internal energy consumption during the multi-step energy transfer processes still resulted in low quantum yields and photolysis efficiency.”*

3. It is reported that near-infrared photolysis is more efficient than green-light and red-light photolysis, but the reason for this is not fully understood.

Response: We acknowledge the reviewer to point out that the mechanism of the high photolytic efficiency remained unclear in our system. Referring to the previous study, the efficiency of upconversion systems depends on the photostability of the energy acceptors. Typically, large π -

conjugated annihilators, such as rubrene, were bleached during the energy transfer processes (ACS Appl. Mater. Interfaces 2018, 10, 12, 9883–9888). Thus, we hypothesized that the photostability of the BODIPY photocage under light irradiation affected its photolytic efficiency.

To verify the hypothesis, we tested the photobleaching of the BODIPY-OH photocage (compound **2**) under 530 nm green light, 625 nm red light (in presence of PtTPBP) and 690 nm NIR light (in presence of Os(btpy)₂²⁺) (100 mW/cm², 0-7 min). The consumption of BODIPY-OH after light irradiation reflected the photobleaching of the photocage. As shown in the below figure (Figure S6 in the revised Supporting Information), BODIPY photocage bleached fastest under green light, slow under red light, and slowest under NIR light. Since the NIR light-triggered photolysis exhibited the highest photolytic yield, it can be concluded that this process increased the photolytic yield by reducing the photobleaching of the prodrugs during photolysis. Notably, a recent publication also reported that the long-wavelength low-energy photons caused less photobleaching of energy acceptor and thus increased the efficiency of a TTET-based photocatalytic system (Nature Communications 14.1 (2023): 1102), which is consistent with our findings.

In the revised manuscript, we added sentences in the Section “Photolysis of BODIPY-based prodrugs via upconversion-like process” (page 10): *“It was reported that the stability of energy acceptors affected the efficiency of upconversion or photochemical reactions.^{24, 25} Thus, we evaluated the photodamage of the BODIPY-OH photocage under 530 nm green light, 625 nm red light (in presence of PtTPBP) and 690 nm NIR light (in presence of Os(btpy)₂²⁺) (100 mW/cm², 0-7 min). As shown in Figure S6, the photocage bleached fastest under green light, slow under red light, and slowest under NIR light, showcasing that utilizing low-energy NIR photons and simplifying energy transfer processes can reduce photodamage of the prodrugs and unexpected relaxation of the excited states.”*

Figure. S6. (A-E) HPLC trace of BODIPY-OH photocage under 530 nm green light, 625 nm red light (with/without PtTPBP) and 690 nm NIR light (with/without Os(btpy)₂²⁺) (100 mW/cm², 0-7 min) (DAD detector, 540 nm). (F) Normalized remaining amount of BODIPY-OH in different groups.

4. Around line 220, it is explained that photolysis does not occur due to low ISC efficiency and short T₁ lifetime without iodine atoms, but this explanation is unreasonable. The ISC efficiency should not be relevant for the photolysis efficiency, since the proposed mechanism does not require ISC for the BODIPY derivative. In addition, the T₁ lifetime is likely to be rather shortened by the heavy atom effect when the iodine atom is modified. This should be evaluated by transient absorption measurements.

Response: We acknowledge the reviewer's comment and suggestion. We agree that our proposed mechanism does not require ISC of the prodrug. Thus, we have deleted the statement accordingly in the revised manuscript (Page 13). To investigate the potential reasons, we conducted transient absorption measurements according to the reviewer's suggestion. It was observed that the triplet state lifetime of BODIPY-Cb (compound **4**) is much longer than that of BODIPY2-Cb (compound **14**), indicating that the iodination prolonged the triplet lifetime of BODIPY. Moreover, BODIPY-Cb displayed a decay trace at 540 nm, which was assigned to the absorption of the triplet state, and the lifetime was determined as 5.274 μs. For BODIPY2-Cb, the triplet lifetime was less than 10 ns. Thus, it can be concluded that the photolysis efficiency depends on the triplet lifetime of the BODIPY prodrugs. The revised manuscript describes the result as "*Furthermore, BODIPY2-Cb prodrug (compound **14**), whose structure is similar to that of BODIPY-Cb but without iodine insertion at the 2- and 6-positions, showed no photolytic yield upon NIR light in the presence of Os(bptpy)₂²⁺ (Figure S16), which can be explained by that the lack of iodine atoms leads to insufficient population and fast decay of T₁ state.²⁹ Moreover, the nanosecond transient absorption spectra (Figure S17) and its decay trace (Figure S18) at 540 nm revealed longer T₁ lifetime of BODIPY-Cb than that of BODIPY2-Cb.*" (Page 13).

Figure. S17. Nanosecond transient absorption spectrum of (A) BODIPY-Cb and (B) BODIPY2-Cb in N_2 -saturated toluene. (Concentration: 10^{-5} M; λ_{ex} = 355 nm).

Figure. S18. Decay trace of BODIPY-Cb at 540 nm at 20 °C. (Concentration: 10^{-5} M; λ_{ex} = 355 nm).

Reviewer #3:

Near-infrared (NIR) light-triggered drug release is preferable for drug delivery and precise disease treatment due to its deep tissue penetration and low phototoxicity. This manuscript by Wang et al developed a new one-step triplet-triplet energy transfer strategy for NIR light-triggered prodrug photolysis, based on Os (II) bromophenyl terpyridine complex as a photosensitizer and a wide range of boron-dipyrromethene (BODIPY)-based prodrugs and bioactive molecules. Moreover, a micellar nanosystem encapsulating both photosensitizer and prodrug has been demonstrated the practicality for in vivo cancer therapy in normoxia aqueous environment. The present work would appeal to a broad readership of the journal, and I support publication after minor revision.

Response: We acknowledge the review's positive evaluation of our work. According to the specific comments, we carefully revised the manuscript. We provided point-by-point responses to the comments as follows:

1. Please provide DFT and TD-DFT calculations to verify the assignment of the S1 and T1 states of Os (II) bromophenyl terpyridine complex in the UV-vis absorption spectrum.

Response: We acknowledge the reviewer's suggestion on DFT and TD-DFT calculation to verify the UV-vis absorption spectrum. We have conducted the calculations and the methodology was added as (page 11 of the revised *Supporting Information*): *“Briefly, time-dependent DFT (TD-DFT) were used for the geometry optimizations with a B3LYP functional. LANL2DZ basis set was used for the iodine and osmium atoms, while 6-31G(d) basis was used for the other atoms. After optimizing the geometry of the ground singlet state, excited singlet state and excited triplet state of BODIPY photocage, Os and BC, the energy level were calculated and recorded. The vertical excitation energies of the singlet states and triplet states were computed at the optimized ground state geometries. All calculations were carried out in the Gaussian16 software package.”*

In the revised manuscript, we added the results of calculations (page 8). *“Based on the phosphorescence emission, the T_1 energy level of $Os(bptpy)_2^{2+}$ was determined as 1.69 eV, which is close to the results of density functional theory (DFT) calculation (Figure S1 and Table S1).²³ The theoretical excitation energies of Os agree well with the experimental spectra (Figure 2A, Table S7). Notably, the S_0-T_1 transition was calculated to be at 679 nm, which fitted the experimental results and verified the S-T transition compatibility of Os.”*

Besides the Os complex, we also calculated the energy level of BC and BODIPY-OH photocage. The results and discussion can be seen on Page 8 as: *“The energy levels of T_1 of BC were calculated as 1.51 eV based on DFT calculations (Figure S2). Moreover, the T_1 energy of the BODIPY-OH photocage was determined as 1.54 eV, which is close to that of BC prodrug, implying that T_1 energy level of the prodrug mainly depends on its photocage moiety (Figure S3).”*

Figure. S1 Optimized excited state geometries and energy levels of $\text{Os}(\text{bptpy})_2^{2+}$.

Table S7. Summarized data for major excitations from TD-DFT calculations of Os.

State	Composition		λ (nm)	Character
			(Energy (eV))	
T_1	HOMO-4 \rightarrow LUMO+1	14.78%	678.91	$^3\text{MLCT}$
	HOMO-3 \rightarrow LUMO	15.93%	(1.8262)	$^3\text{MLCT}$
	HOMO-1 \rightarrow LUMO	20.82%		$^3\text{MLCT}/^3\text{ILCT}$
	HOMO-1 \rightarrow LUMO+1	12.49%		$^3\text{MLCT}/^3\text{ILCT}$
	HOMO \rightarrow LUMO	13.57%		$^3\text{MLCT}/^3\text{ILCT}$
	HOMO \rightarrow LUMO+1	19.17%		$^3\text{MLCT}/^3\text{ILCT}$
T_2	HOMO-4 \rightarrow LUMO	13.64%	654.04	$^3\text{MLCT}$
	HOMO-3 \rightarrow LUMO+1	12.92%	(1.8957)	$^3\text{MLCT}$
	HOMO-1 \rightarrow LUMO	14.28%		$^3\text{MLCT}/^3\text{ILCT}$
	HOMO-1 \rightarrow LUMO+1	20.21%		$^3\text{MLCT}/^3\text{ILCT}$
	HOMO \rightarrow LUMO	21.61%		$^3\text{MLCT}/^3\text{ILCT}$
	HOMO \rightarrow LUMO+1	13.36%		$^3\text{MLCT}/^3\text{ILCT}$
T_3	HOMO-2 \rightarrow LUMO	93.14%	585.85 (2.1163)	$^3\text{MLCT}$
S_1	HOMO-4 \rightarrow LUMO	13.78%	556.65 (2.2273)	$^3\text{MLCT}$
	HOMO-3 \rightarrow LUMO+1	13.20%	$f=0.0005$	$^3\text{MLCT}$
	HOMO-1 \rightarrow LUMO	15.13%		$^1\text{MLCT}/^1\text{ILCT}$
	HOMO-1 \rightarrow LUMO+1	20.66%		$^1\text{MLCT}/^1\text{ILCT}$
	HOMO \rightarrow LUMO	23.25%		$^1\text{MLCT}/^1\text{ILCT}$
	HOMO \rightarrow LUMO+1	13.48%		$^1\text{MLCT}/^1\text{ILCT}$

Figure. S2. Optimized excited state geometries and energy levels of BODIPY-Cb.

Figure. S3. Optimized excited state geometries and energy levels of BODIPY-OH.

2. Please explain how to determine the amount of released Chlorambucil drug in vivo.

Response: We acknowledge the review's suggestion. We agree with the idea to determine the amount of released Chlorambucil drug *in vivo*. The method for verifying *in vivo* prodrug activation has been well-established based on LC/MS/MS (Journal of Pharmaceutical and Biomedical Analysis 99 (2014): 74-78). The procedure that we used to determine the amount of released chlorambucil *in vivo* was described as (Page 18 of the *Supporting Information*): "LC/MS/MS was used to quantify the prodrug activation in tumor. Briefly, the Os/BC NPs were injected into the tail vein of tumor-bearing mice (tumor volume at around 200 mm³, n=3), of which the dosage was the same as the therapeutic settings. For the light irradiation group, NIR light (690 nm, 300 mW/cm², 10 min) were applied on the tumor area 24 h post-injection. Then the mice were euthanized. Tumors and major organs were collected, weighed, and homogenized. The prodrug (BC) and free drug (chlorambucil) were extracted by acetonitrile from the tissue homogenates, followed by the centrifugation at 12,000 rpm for 10 min, and the

supernatant was collected and filtered through 220 nm filter before injecting into LC/MS/MS system (Agilent 1290/AB SCIEX 3200 QTRAP). The concentrations of free Cb and BC in tissues were determined by multiple reaction monitoring (MRM) of ions.”

As a result, we successfully observed the *in vivo* activation of BC in the tumor tissues. The descriptions are added in the revised manuscript (page 22). “For quantification of *in vivo* prodrug activation, LC/MS/MS was used to quantify the prodrug and free drug in tissues. It was found that both BC and Cb exhibited obvious relative abundance in LC/MS/MS chromatograms at low concentrations (1-400 ng/mL and 1-100 ng/mL, respectively) (Figure S23 and S24) At 24 h after the *i.v.* injection of OS/BC NPs, it was observed that the BC prodrug distributed in tumor tissues and the major organs (Figure 6H). Obvious BC consumption was observed in tumors after NIR light irradiation, as well as the release of free Cb (Figure 6I). Notably, since the light irradiation was only performed at the tumor area, no free Cb was observed in the other tissues, indicating the excellent tumor specificity of our treatment.”

Figure. S23. Linear fitting of the concentrations of (A) BC and (B) Cb and analyte peak areas measured by LC/MS/MS system.

Figure. S24. Representative MRM chromatograms of tumor tissues 24 h post-treatments (i.v. injection of Os/BC NPs, with/without NIR light irradiation at tumors; light irradiation: 690 nm, 300 mW/cm², 10 min).

Figure 6. (A) Representative IVIS fluorescence images of the mice after injection of free DiR and DiR NPs within 24 h (n = 3). Red dashed circles indicate tumor areas. (B) Quantitative analysis of biodistribution in major organs and tumor determined by IVIS. Tu, He, Lu, Sp, Li, and Ki represent tumor, heart, lung, spleen, liver, and kidney, respectively. *** p < 0.005. (C) Schematic illustration of the treatment schedule. (D) Photograph of a mouse irradiated with NIR light at the tumor site. (E) Tumor volume of each group (n = 5). (F) Photograph of tumors resected at Day 13 after different treatments. (G) Body weight of each group. Concentration of (H) BC and (I) Cb in tumors and organs at 24 h after intravenous administration of Os/BC NPs with/without light irradiation on tumors (n = 3). (J) Representative H&E staining of tumor sections of different treatment groups. Scale bar: 200 μ m. Light irradiation: 690 nm, 300 mW/cm², 10 min.

Reviewers' Comments:

Reviewer #2:

Remarks to the Author:

I found that the authors carefully addressed all my concerns. Now I support the acceptance of this manuscript in Nature Communications.

Reviewer #3:

Remarks to the Author:

The revised manuscript has well addressed the problems raised by the reviewer, I recommend acceptance as it is.

Reviewer #4:

Remarks to the Author:

About the calculations:

The authors provided the calculations as suggested by the Reviewer #3, and the computational details are "Briefly, time-dependent DFT (TD-DFT) were used for the geometry optimizations with a B3LYP functional. LANL2DZ basis set was used for the iodine and osmium atoms, while 6-31G(d) basis was used for the other atoms. After optimizing the geometry of the ground singlet state, excited singlet state and excited triplet state of BODIPY photocage, Os and BC, the energy level were calculated and recorded. The vertical excitation energies of the singlet states and triplet states were computed at the optimized ground state geometries. All calculations were carried out in the Gaussian16 software package", shown in the Response for Reviewer #3. However, the sentence "The vertical excitation energies of the singlet states and triplet states were computed at the optimized ground state geometries" was missed in page 11 of the revised Supporting Information for TD-DFT calculations. The related discussions were added in the revised manuscript "Based on the phosphorescence emission, the T1 energy level of Os(bptpy)₂²⁺ was determined as 1.69 eV, which is close to the results of density functional theory (DFT) calculation (Figure S1 and Table S1). The theoretical excitation energies of Os agree well with the experimental spectra (Figure 2A, Table S7). Notably, the S0-T1 transition was calculated to be at 679 nm, which fitted the experimental results and verified the S-T transition compatibility of Os. The energy levels of T1 of BC were calculated as 1.51 eV based on DFT calculations (Figure S2). Moreover, the T1 energy of the BODIPY-OH photocage was determined as 1.54 eV, which is close to that of BC prodrug, implying that T1 energy level of the prodrug mainly depends on its photocage moiety (Figure S3)." In my opinion, the above expressions are not correct. The ground state geometries are optimized by DFT (not the TD-DFT), while the vertical excitation energies of the singlet states and triplet states are computed by TD-DFT at the optimized ground state geometries. In Figure S1-S3, the optimized excited state geometries and energy levels should also be performed by TD-DFT. Thus, the words "density functional theory (DFT) calculation" in the main text should be "time-dependent density functional theory (DFT) calculation".

About the energy transfer mechanisms:

The Reviewer #1 mentioned that S-T transition of Os complexes is weak. Can the authors provide the oscillator strength (f) for S-T transition by TD-DFT? For metal-organic complexes, the strong S-T absorption means that the spin-orbit coupling between singlet and triplet excited states should be large enough, so that the triplet state can borrow some oscillator strengths from singlet state. But the calculated f for S1 is very weak. Moreover, the spin-orbit coupling for T1→S0 is also improved, as a result, the triplet lifetime is not too long, which goes against the energy transfer process and the final photolysis efficiency. The authors should provide some feasible strategies for further development.

Reviewer #5:

Remarks to the Author:

[Note from the Editor: Reviewer #5 was asked to assess the response given to Reviewer #1.]

In this work, Wang and co-workers developed a new one-step energy transfer strategy for NIR light-triggered prodrug photolysis by combining Osmium-based photosensitizer and BODIPY-based prodrug. They further demonstrated the application of the NIR light-cleavable prodrugs system for photo-activated cancer therapy at the cellular level and in tumor-bearing living mice. Although the photo-controlled prodrug activation has been widely reported, the prodrug photolysis with high yields under low-irradiance NIR light is still interesting. And I would like to praise the authors for their careful and detailed responses to the reviewers' previous comments. However, I hope they can enhance the paper's quality by addressing the following questions and suggestions.

Some remaining major concerns:

1. The author stated that this strategy could be used for the controlled activation of a wide range of drugs and bioactive molecules, including some anti-cancer drugs, anti-inflammation drugs, anesthesia agent, and biogenic amines. However, as shown in the cell growth and tumor suppression experiments, the Osmium-based photosensitizer alone showed some phototoxicity. Will this phototoxicity cause side effects to targeted tissue, especially when they are used for anti-inflammation or anesthesia application?
2. Please clarify the superiority of this strategy over other commonly used photo-based therapies, such as PTT or PDT-based cancer treatment, which can kill tumors effectively. From the in vivo anti-tumor study, we could see that the tumors were not completely suppressed after two rounds of NP administration and light irradiation.
3. In addition, multiple biomarker-triggered prodrug release as well as the NIR-II light- or ultrasound-activated prodrug release have been reported previously. A discussion about the achievement made by this work when compared to these strategies is required.
4. The hematology parameters are also very important indicators to evaluate in vivo biocompatibility, thus complete blood panel tests were suggested to indicate the biosafety of Os/BC NPs. In addition, as one of the main advantages of photocleavable prodrugs is reducing systemic toxicity, please experimentally validated the high safety of Os/BC NPs over free drugs.

Minor concerns

1. In line 259, it is mentioned that the absorption spectra of Os/BC NPs indicated successful encapsulation of Os(btpy)₂²⁺ and BODIPY-Cb in the nanoparticles. Please give the encapsulation efficiency of Os(btpy)₂²⁺ and BODIPY-Cb in the nanomicelles. Did the author optimize the loading ratio of them to achieve the high-efficient prodrug photolysis?
2. Please also provide the release behavior of Os(btpy)₂²⁺ and BODIPY-Cb from nanomicelles in physiological conditions. As the photoirradiation was conducted 24 h after the injection of NPs into mice, will the pre-release of photosensitizer and prodrug in tissue affect the photolysis as the photocleavage reaction was significantly quenched in the presence of oxygen?
3. The photolytic yields of prodrug 4-13 vary from 46% to 87%, please give the reason behind this.
4. There is grammar issue in line 339. Please double-check it.
5. For Figure 6, the order of data and its corresponding description are required to be adjusted to make them easier to follow. For example, it is better to give the prodrug activation data first.

REVIEWER COMMENTS

Reviewer #2 (Remarks to the Author):

I found that the authors carefully addressed all my concerns. Now I support the acceptance of this manuscript in Nature Communications.

Response: We appreciate the reviewer's positive evaluation of our efforts.

Reviewer #3 (Remarks to the Author):

The revised manuscript has well addressed the problems raised by the reviewer; I recommend acceptance as it is.

Response: We appreciate the reviewer's positive evaluation of our efforts.

Reviewer #4 (Remarks to the Author):

About the calculations:

The authors provided the calculations as suggested by the Reviewer #3, and the computational details are “Briefly, time-dependent DFT (TD-DFT) were used for the geometry optimizations with a B3LYP functional. LANL2DZ basis set was used for the iodine and osmium atoms, while 6-31G(d) basis was used for the other atoms. After optimizing the geometry of the ground singlet state, excited singlet state and excited triplet state of BODIPY photocage, Os and BC, the energy level were calculated and recorded. The vertical excitation energies of the singlet states and triplet states were computed at the optimized ground state geometries. All calculations were carried out in the Gaussian16 software package”, shown in the Response for Reviewer #3. However, the sentence “The vertical excitation energies of the singlet states and triplet states were computed at the optimized ground state geometries” was missed in page 11 of the revised Supporting Information for TD-DFT calculations.

Response: We acknowledge the reviewer for careful reviewing and suggestions. We apologize for this mistake. We have added the corresponding sentence in the revised *Supporting Information as below:*

“DFT and TD-DFT calculations

Briefly, density functional theory (DFT) and time-dependent DFT (TD-DFT) were used with a B3LYP functional for geometry optimizations and energy computation, respectively. The LANL2DZ basis set was used for the iodine and osmium atoms, while the 6-31G(d) basis set was used for the other atoms. After optimizing the geometry of the ground singlet state, excited singlet state and excited triplet state of BODIPY photocage, Os and BC, the energy level were calculated and recorded. The vertical excitation energies of the singlet states and triplet states were computed at the optimized ground state geometries. All calculations were carried out in the Gaussian16 software package.” (page 13, Supporting Information)

The related discussions were added in the revised manuscript “Based on the phosphorescence emission, the T₁ energy level of Os(bptpy)₂²⁺ was determined as 1.69 eV, which is close to the results of density functional theory (DFT) calculation (Figure S1 and Table S1). The theoretical excitation energies of Os agree well with the experimental spectra (Figure 2A, Table S7). Notably, the S₀-T₁ transition was calculated to be at 679 nm, which fitted the experimental results and verified the S-T transition compatibility of Os. The energy levels of T₁ of BC were calculated as 1.51 eV based on DFT calculations (Figure S2). Moreover, the T₁ energy of the BODIPY-OH photocage was determined as 1.54 eV, which is close to that of BC prodrug, implying that T₁ energy level of the prodrug mainly depends on its photocage moiety (Figure S3).” In my opinion, the above expressions are not correct. The ground state geometries are optimized by DFT (not the TD-DFT), while the vertical excitation energies of the singlet states and triplet states are computed by TD-DFT at the optimized ground state geometries. In Figure S1-S3, the optimized excited state geometries and energy levels should also be performed by TD-DFT. Thus, the words “density functional theory (DFT) calculation” in the main text should be “time-dependent density functional theory (DFT) calculation”.

Response: We appreciate the reviewer for pointing out these mistakes. We agree with the reviewer’s comment that we should distinguish “DFT” (for geometries optimizations) and “TD-DFT” (for energy calculation of exciting states). In the

revised version of manuscript, we have revised the corresponding sentences and figures as follow:

“Based on the phosphorescence emission, the T_1 energy level of $Os(bptpy)_2^{2+}$ was determined as 1.69 eV, which is close to the result of time-dependent density functional theory (TD-DFT) calculation (Figure S1 and Table S1).” (page 8, line 135)

“The T_1 energy level of BC was calculated as 1.51 eV based on TD-DFT calculations (Figure S2).” (page 8, line 140)

About the energy transfer mechanisms:

The Reviewer #1 mentioned that S-T transition of Os complexes is weak. Can the authors provide the oscillator strength (f) for S-T transition by TD-DFT? For metal-organic complexes, the strong S-T absorption means that the spin-orbit coupling between singlet and triplet excited states should be large enough, so that the triplet state can borrow some oscillator strengths from singlet state. But the calculated f for S_1 is very weak. Moreover, the spin-orbit coupling for $T_1 \rightarrow S_0$ is also improved, as a result, the triplet lifetime is not too long, which goes against the energy transfer process and the final photolysis efficiency. The authors should provide some feasible strategies for further development.

Response: We acknowledge the reviewer’s suggestions and comments on the TD-DFT calculations. We agree that the spin-orbit coupling intensity can be predicted by calculating the oscillator strength. Previously, we used B3LYP functional and LANL2DZ basis set, of which the results did not give enough details about the oscillator strength of the triplet states of Os. Here, Orca 5.0.3 software was used to calculate the oscillator strength of the triplet states at the optimized geometry of the ground state by using B3LYP functional and DKH-def2-TZVP(-f) basis set (SARC-DKH-TZVP for Os) (*Wiley Interdisciplinary Reviews: Computational Molecular Science* 2018, 8, 1, e1327). The result indicated that the triplet state of Os ($\lambda = 673.2$ nm) borrows oscillator strength mainly from the S_5 state (highlighted in Table S7). The oscillator strength of S_5 decreases from 0.4133 to 0.3980 after considering the

spin orbit coupling (SOC) effect. The SOC corrected absorption spectrum (Figure S46) also proves the S-T absorption of Os, which is consistent with the experimental spectrum ($\lambda=678$ nm, Figure 2A). All these results indicated that the strong spin-orbit coupling of the Os complex enables the efficient S-T transition by NIR light excitation.

Table S7. Summarized data of the calculated excitation wavelengths and oscillator strength of Os(bptpy)₂²⁺ with/without considering SOC effects.

B3LYP			SOC corrected					
Wavelength/nm	states	fosc	wavelength/nm	states	fosc	wavelength/nm	states	fosc
583.5	S ₁	0.000345085	752.5	T ₁	0.000486178	523.3	S₅	0.397998611
565.1	S ₂	0.000000003	751.8	T ₁	0.000683882	520.3		0.003790499
548.2	S ₃	0.011971852	739.7	T ₁	0.000055331	519		0.001971716
545.5	S ₄	0.011272397	730.4	T ₂	0	517.2		0.000008343
538.6	S₅	0.413314929	714.0	T ₂	0.001236076	515.8		0.000013032
519.5	S ₆	0.01502482	713.4	T ₂	0.001366506	512.5		0.007068029
519.2	S ₇	0.015605847	673.2	T₄	0.068508116	499.2		0.007251533
511.2	S ₈	0.000000009	645.3		0.00000189	499.2		0.035463615
494.2	S ₉	0.007128747	614.6		0.000009985	497		0.00000638
494	S ₁₀	0.006981026	614		0.000021391	490.7		0.00000075
477.5	S ₁₁	0.239702248	608.8		0.000005048	489		0.000197688
451.1	S ₁₂	0.000000003	606.5		0.000000006	488.4		0.000169484
433.7	S ₁₃	0.000001893	601.9		0.001717431	488.4		0.000000001
431.3	S ₁₄	0.000000002	599.9		0.001798423	486.7		0.001747015
427.4	S ₁₅	0.00590167	587.1		0.000000206	479.1		0.001749633
716.1	T ₁		584.3		0.010773528	479.1		0.000000001
687.6	T ₂		582.2		0.00157331	477		0.000000029
612.2	T ₃		581		0.001636338	475.8		0.001158916
611	T ₄		579.9		0.000065336	475.8		0.001074647
609.4	T ₅		571.6		0.001450407	475.5		0.002250997
589.5	T ₆		564.1		0.000000584	473.9		0.00200824
559.5	T ₇		562.2		0.001475821	473.8		0.146891133
559.5	T ₈		561.5		0.002328637	463.8		0.000042018
529.1	T ₉		557.8		0.00523346	444.1		0.00000118
529	T ₁₀		557.2		0.004424795	433.9		0.005177808
506.3	T ₁₁		553.5	S ₁	0.000377127	433		0.002197719
506.1	T ₁₂		553.1		0.00000036	432.5		0.00107536
482	T ₁₃		540.2	S ₂	0.000000004	429.8		0.000000004
479.9	T ₁₄		526.6	S ₃	0.009343234	426.3		0.00289337
439.5	T ₁₅		525	S ₄	0.006782797	422.6		0.397998611

Figure S46. Normalized calculated absorption spectrum of Os(btpy)₂²⁺.

The reviewer mentioned that “the authors should provide some feasible strategies for further developments of Os photosensitizers.” Indeed, as emerging photosensitizers, the Os complexes exhibited shortcomings, such as the relatively short triplet lifetime. Overcoming this problem has attracted many research interests. For example, Kimizuka et.al. reported that the triplet lifetime of Os photosensitizers can be prolonged by covalently conjugating moieties with large π -conjugation structure and long triplet lifetime, such as perylene. This strategy was summarized as enhancing the intramolecular triplet energy transfer (IMET) between the metal centre and ligand by excited-state thermal equilibrium. (*Angew. Chem. Int. Ed.* 2019, 58, 49, 17827-17833; *Inorg. Chem.* 2022, 61, 16, 5982-5990.) Besides, other strategies, such as discovering S-T photosensitizers with new metal centers (*J. Am. Chem. Soc.* 2021, 143, 3, 1651-1663), modulating chromophore environment (*J. Mater. Chem. C*, 2022, 10, 13747-13752), and stabilizing triplet states by non-covalent interactions (*Dalton Trans.* 20, 2009, 3980-3987), will properly be workable to promote the photophysical performance of the STPSs and finally, increase the efficiency of the energy-transfer based photolysis reaction in the future.

The above discussions were added in the discussion part of the revised manuscript as “*The triplet lifetime of PS is important in the energy transfer process between the STPSs and prodrugs. It was reported that long-triplet-lifetime Os photosensitizers can be prepared by covalently conjugating moieties with large π -conjugation structure and long triplet lifetime, such as perylene, which enhanced the intramolecular triplet*

energy transfer between the metal centre and ligand by excited-state thermal equilibrium.^{40, 41} Other strategies, such as discovering STPSs with new metal centres,⁴² modulating chromophore environment,⁴³ and stabilizing triplet states by non-covalent interactions,⁴⁴ will properly be workable to promote the photophysical performance of the STPSs and, finally, increase the efficiency of the energy transfer-based photolysis reactions in the future.” (page 24, line 441)

Reviewer #5 (Remarks to the Author):

In this work, Wang and co-workers developed a new one-step energy transfer strategy for NIR light-triggered prodrug photolysis by combining Osmium-based photosensitizer and BODIPY-based prodrug. They further demonstrated the application of the NIR light-cleavable prodrugs system for photo-activated cancer therapy at the cellular level and in tumor-bearing living mice. Although the photo-controlled prodrug activation has been widely reported, the prodrug photolysis with high yields under low-irradiance NIR light is still interesting. And I would like to praise the authors for their careful and detailed responses to the reviewers' previous comments. However, I hope they can enhance the paper's quality by addressing the following questions and suggestions.

Response: We acknowledge the reviewer's comments and appreciation of our previous revision. For the questions and suggestions, we conducted additional experiments and provided point-by-point responses as below:

Some remaining major concerns:

1. The author stated that this strategy could be used for the controlled activation of a wide range of drugs and bioactive molecules, including some anti-cancer drugs, anti-inflammation drugs, anesthesia agent, and biogenic amines. However, as shown in the cell growth and tumor suppression experiments, the Osmium-based photosensitizer alone showed some phototoxicity. Will this phototoxicity cause side effects to targeted tissue, especially when they are used for anti-inflammation or anesthesia application?

Response: We acknowledge the reviewer's concern on the phototoxicity and side effects. It was observed that the phototoxicity of osmium-based photosensitizer is not obvious until the concentration was relatively high (62.6 % cells were alive at 50 μM) (Figure 5C). As compared to the light-activated prodrug group (Os/BC NPs), it can be concluded that the cell viability was mainly affected by the prodrug activation (10.9 % cells alive at 50 μM). Notably, the Os/BC NPs with lower concentrations also

showed good anti-proliferative effects, while the Os NPs alone exhibited limited phototoxicity at the same concentrations.

Figure 5. (C) Cytotoxicity of Os NPs, BC NPs and Os/BC NPs with light irradiation against HeLa cells (n = 5). (690 nm, 100 mW/cm², 30 min)

For further evaluation, we investigated the singlet oxygen generation by SOSG assay. In Figure S22, we have observed that the ¹O₂ generation by Os(btpy)₂²⁺ under 690 nm light was much lower than methyl blue (MB), a commercially available photosensitizing agent, indicated that Os(btpy)₂²⁺ alone is not potentially toxic as compared to the traditional photosensitizer. Based on this result, we added a sentence in the revised manuscript (page 17, line 307) as: “*The low phototoxicity of Os photosensitizer alone can be explained by the limited singlet oxygen generation capability as compared to methyl blue (Figure S22).*”

Figure S22. Singlet oxygen generation of methyl blue (MB) (A) and Os NPs (B), detected by SOSG assay. (C) Time-dependent F/F₀-1 values at 525 nm under NIR light irradiation. (Concentration: 1 µM; light irradiation: 690 nm, 10 mW/cm², 0-10 s)

We then investigated the potentials of our system for anti-inflammation applications. BODIPY-indomethacin (BI) and BODIPY-naproxen (BN) (prodrug 6-7) which exhibited high photolytic yields and anti-inflammation effects were chose. After constructing Os/BI NPs and Os/BN NPs, the anti-inflammation effects were tested by Griess reagent in lipopolysaccharide (LPS)-activated RAW264.7 macrophages. Nitric oxide (NO) is an inflammatory mediator, of which the content reveals the progression of inflammation (*Adv. Healthcare Mater.* 2023, 2301394). Increased NO production was observed after LPS-mediated activation, which was dose-dependently suppressed by Os/BI NPs or Os/BN NPs with light irradiation (Figure S23A and C). These results verified the anti-inflammatory effects of light-triggered release of clinical anti-inflammatory drugs, such as indomethacin and naproxen, with negligible cytotoxicity (Figure S23B and D).

Figure S23. Light-triggered release of indomethacin or naproxen reduced LPS-induced inflammatory responses of RAW 264.7 cells. (A) The NO content and (B) relative cell viability of RAW 264.7 cells treated with LPS and Os/BI NPs with/without light irradiation. (C) The NO content and (D) relative cell viability of RAW 264.7 cells treated with LPS and Os/BN NPs with/without light irradiation. *p < 0.05, **p < 0.01, ***p < 0.001.

Related descriptions can be found in the revised manuscript. *“For the other prodrugs, such as BODIPY-indomethacin (BI) and BODIPY-naproxen (BN) (prodrug 6-7), similar nanoparticles (Os/BI NPs and Os/BN NPs) were prepared, and their anti-inflammation effects were tested by Griess reagent in lipopolysaccharide (LPS)-activated RAW264.7 macrophages. Nitric oxide (NO) is an inflammatory mediator, of which the content reveals the progression of inflammation.³⁷ Increased NO production was observed after LPS-mediated activation, which was dose-dependently suppressed by Os/BI NPs or Os/BN NPs with light irradiation (Figure S23). These results verified the anti-inflammatory effects of light-triggered release of clinical anti-inflammatory drugs, indomethacin, and naproxen, with negligible cytotoxicity.”* (page 17, line 316)

The related methodology is added in the *Supporting Information* as below:

“Nitric Oxide Content detection in RAW 264.7 cells

The concentration of nitric oxide of LPS-stimulated RAW 264.7 cells was determined by Griess reagent. RAW 264.7 cells in complete DMEM medium were treated with LPS (20 ng/mL) and the formulations (PBS, Os/BI NPs, Os/BN NPs) at different concentrations (1-5 μ M on basis of prodrugs). For light-triggered groups, NIR light (690 nm, 100 mW/cm², 30 min) was applied after 4 h incubation. After 24 h, 100 μ L cultural supernatant was collected and reacted with an equal volume of Griess reagent (40 mg/mL in H₂O). The absorbance was measured at 540 nm to determine the NO concentrations.” (page 17, *Supporting Information*)

In all, we acknowledge the reviewer’s concern on the phototoxicity and side effects. Based on the above results, it can be concluded that the Os photosensitizer exhibits limited phototoxicity even at relatively high concentrations. Its phototoxicity should be much lower as compared to conventional photosensitizers, like MB, due to the relatively low ¹O₂ production under light irradiation. The anti-inflammation effects were tested in LPS-activated RAW264.7 macrophages. Our system, which released

clinical anti-inflammatory drugs, indomethacin and naproxen, upon light irradiation, suppressed inflammation with negligible cytotoxicity. Thus, with careful dosage controlling, it can be anticipated that our system can be used for safe applications in anti-inflammation, anesthesia, etc., with precise control by NIR light.

2. Please clarify the superiority of this strategy over other commonly used photo-based therapies, such as PTT or PDT-based cancer treatment, which can kill tumors effectively. From the *in vivo* anti-tumor study, we could see that the tumors were not completely suppressed after two rounds of NP administration and light irradiation.

Response: We acknowledge the reviewer's comment about comparing our system (light-triggered prodrug activation) and traditional phototherapies (PDT and PTT). Based on PDT or PTT, tumor cells can be killed by the toxic reactive oxygen species (ROS) or local heating with laser irradiation, which allows good flexibility of operation and precise spatiotemporal control. However, limitations exist, such as the oxygen-relying feature of PDT, the high-power laser requirement for PTT, and the potential resistance to single therapy (*Chem. Soc. Rev.* 2016, 45, 6597-6626; *Adv. Sci.* 2021, 8, 3, 2002504). As compared, the strategy introduced in this study used light to activate prodrugs and enabled the accurately controllable release of drugs with different pharmacological functions. Thus, the therapeutic effect of such prodrug systems can be more diverse than the traditional PDT and PTT, and thus can adapt to different treatment requirements. Also, the photolysis reaction works well in hypoxia environments and the required light irradiance is relatively low (100 mW/cm²).

The reviewer pointed out that "the tumors were not completely suppressed after two rounds of NP administration and light irradiation". It was due to the limited therapeutic effect of chlorambucil, the model drug in this study. We would like to emphasize here that we only involved the chlorambucil prodrug as therapeutic agent for the *in vivo* tests. As the proof of concept, the results of this study verified the therapeutic effect of light-activated prodrug in tumors. We can expect that the

efficacy can be improved by discovering more potent anti-cancer prodrugs or combining PDT/PTT agents with the photoactivatable prodrugs.

Some sentences in the above discussions were added in the revised manuscript to emphasize the advancements of our system: *“The strategy of this study that uses light to activate prodrugs has enabled the accurately controllable release of drugs with different pharmacological functions. Thus, the therapeutic effect of such prodrug systems can be diverse to adapt to different therapeutic requirements. Besides, such photolysis reaction can work well in hypoxia environments and the required excitation power density is relatively low (100 mW/cm²).”* (page 25, line 453)

3. In addition, multiple biomarker-triggered prodrug release as well as the NIR-II light- or ultrasound-activated prodrug release have been reported previously. A discussion about the achievement made by this work when compared to these strategies is required.

Response: We appreciate the reviewer’s suggestion. We agree with the reviewer’s statement that multiple biomarker-triggered prodrug release as well as NIR-II light- or ultrasound-activated prodrug release have been reported (*Chem. Soc. Rev.*, 2023, 52, 879-920; *Adv. Mater.* 2023, 2300048; *Sci. Adv.* 2023, eadg5964). It should be noted that the current strategies for constructing such prodrugs are still quite limited. For example, biomarker-triggerable prodrugs usually contains specific enzyme-responsive sequences or redox-responsive linkers. Besides, the current NIR-II light- or ultrasound-activatable prodrugs are mainly limited in the platinum-based drugs or the redox/thermos-responsive structures with the combination of PDT/PTT. In this study, we have developed a novel strategy that was applicable for a wide range of drugs, including many drugs with carboxylic groups or with amino groups, which will simplify and expand the methodology for developing photoactivatable prodrugs. Moreover, the energy transfer-based photolysis reaction in this study is highly modular, which means that the wavelength of light can be easily modulated by tuning

the structures and energy levels of photosensitizers. It can be anticipated that NIR-II light can be involved based on the further development of novel photosensitizers that initiates the energy transfer process with NIR-II photon, which will be the future developments of our strategy. Owing to these discussions, we modified the discussions in the manuscript as “*In the future, further developments of this strategy can be anticipated by developing photosensitizers that exhibits longer excitation wavelengths (such as NIR-II light) with deep penetration, more efficient S-T transition capability, longer triplet state lifetime, and more satisfying biocompatibility.*” (page 25, line 462)

4. The hematology parameters are also very important indicators to evaluate *in vivo* biocompatibility, thus complete blood panel tests were suggested to indicate the biosafety of Os/BC NPs. In addition, as one of the main advantages of photocleavable prodrugs is reducing systemic toxicity, please experimentally validated the high safety of Os/BC NPs over free drugs.

Response: We agree with the review’s suggestion that the hematology parameters are useful indicators to evaluate biocompatibility. In the revised manuscript, we have added the serum biochemistry results of the tumor-bearing mice receiving various treatments (PBS, free Cb (8 mg/kg), Os NPs + *hν*, BC NPs + *hν*, and Os/BC NPs ± *hν* (8 mg/kg of Cb)) twice within 14 days. Serum urea (UREA), creatinine (CREA), alanine aminotransferase (ALT) and aspartate aminotransferase (AST) were tested. It was observed ALT in the serum of free Cb-treated mice increased, indicated that of systemic administration of Cb exhibited systemic toxicity.

Figure S29. Serum biochemistry (A. aminotransferase (ALT); B. aspartate aminotransferase (AST); C. urea and D. creatinine (CREA)) of the mice treated with different formulations twice within 14 days. * $p < 0.05$.

The above results are described in the revised manuscript in page 22, line 408: “*Urea, creatinine, alanine aminotransferase (ALT) and aspartate aminotransferase (AST) in serum were determined and no systemic toxicity was observed except for the free Cb-treated group (Figure S29).*”

For the complete blood panel tests, we apologize that we currently do not have suitable instruments for analyzing the mouse blood cells. Alternatively, we have investigated the hemolysis by treating the mouse red blood cells (RBCs) with free Cb and the Os/BC NPs at different concentrations. It was observed that free Cb caused obvious hemolysis with a dose-dependent manner, which is corresponding to the previous toxicology study of Cb resulted in anemia of patients (*Bioconjugate. Chem.* 2014, 25, 11, 2046-2054). As compared, the prodrug nanoparticles (Os/BC NPs) showed significantly reduced hemolysis at the same concentrations of free Cb (Figure S30A).

To further demonstrate the high safety of prodrug nanoparticles over free drug, we investigated the genotoxicity, which is commonly found after the treatments of alkylating agents, such as Cb (*Mutagenesis*, 8.4 (1993): 373-375). In brief, bone marrow cells were collected and stained by Giemsa solution to determine the micronuclei ratio of bone marrow erythrocytes, of which the results have shown the highest genotoxicity of free Cb treatments compared to all the other groups (Figure S30B and C).

Figure S30. (A) Hemolysis test of free Cb and Os/BC NPs at different concentrations. (B) Number of micronuclei in bone marrow cells after different treatments. (C) Representative image of the bone marrow cells of free Cb-treated mice. Black arrows indicate micronuclei.

The descriptions of the above results are added in the manuscript as: “Besides, as shown in Figure S30A, the free Cb treatment resulted in hemolysis, while no hemolysis (hemolysis ratio < 5%) was observed after the nanoparticle treatment. The genotoxicity was evaluated by micronucleus assay and the bone marrow cell micronucleus number in the free Cb-treated mice was much higher than that of the other groups (Figure S30B and C). All these results indicated that the light-activatable prodrug nanoparticles exhibited less toxicity than the systemic administrated Cb.” (page 23, line 411)

The related methodology of hemolytic test and micronucleus assay are added in the Supporting Information as below:

“Hemolytic test

The hemolytic test was conducted by adding free Cb or Os/BC NPs at different concentrations (1-10 μM, on basis of Cb, diluted in PBS) into red blood cells (RBC) suspension, followed by the 2 h incubation at 37 °C. PBS, the negative control, and distilled water, the positive control, were added meanwhile. After incubation, mixtures were centrifuged at 3000 g for 10 min. The hemolysis was determined by the absorbance at 542 nm.” (page 21, Supporting Information)

“Micronucleus assay

Micronucleus assay was conducted according to the previous study. After the treatments with different formulations, bone marrow cells from femurs of the mice were collected and sectioned. After fixation by methanol for 10 min, the cells were stained with 10% Giemsa for 30 min and polychromatic erythrocytes are scored for the incidence of micronuclei (per 500 cells of each mouse).” (page 22, Supporting Information)

Minor concerns

1. In line 259, it is mentioned that the absorption spectra of Os/BC NPs indicated successful encapsulation of Os(btpy)₂²⁺ and BODIPY-Cb in the nanoparticles. Please give the encapsulation efficiency of Os(btpy)₂²⁺ and BODIPY-Cb in the nanomicelles. Did the author optimize the loading ratio of them to achieve the high-efficient prodrug photolysis?

Response: We appreciate the reviewer’s comments. It should be pointed out that we have provided the formulation of Os/BC NPs in Table S5. The encapsulation efficiency of Os(btpy)₂²⁺ and BODIPY-Cb in the nanoparticles were determined as 29.43% and 55.79%, respectively.

The formulation has been carefully optimized, and we apologize for not describing the details before. The optimization was conducted by considering size, PDI, and photolytic efficiency of prodrug at different feeding ratios of the cargoes (Os and BC) to the polymer (PLA-mPEG). First, it was found that the size and PDI greatly increased while feeding more than 0.75% (w/w) Os (Figure S19A), indicating the limit of photosensitizer loading. Then BC at different weight ratios were loaded, and the amount of BC had no influence on the size of nanoparticles (Figure S19B). The photolytic efficiency of prodrug was determined after NIR light irradiation (690 nm, 100 mW/cm², 30 min), and 0.5% BC (total 1.25% cargoes (w/w) in nanoparticles)

was identified as an optimal ratio with relatively high photolytic efficiency (Figure S19C). This formulation can be scaled up and the detail can be seen in Table S5.

Figure S19. Size, PDI of PLA-PEG NPs containing different contents of Os (A) and BC (B). (C) Uncleaved BC percentages of different Os/BC NPs (0.75% Os (w/w) with different feeding ratios of BC under NIR light (690 nm, 100 mW/cm², 30 min).

To emphasize the above additional information, we added the following sentences in the revised manuscript as “*The ratios of photosensitizer (Os) and prodrug (BC) were optimized by feeding different amounts of cargos and recording the size, PDI, and photolytic yields of the prodrug (Figure S19). As shown in Table S5, the optimized formulation of Os/BC NP has the feeding ratio of 0.75% Os (w/w) and 0.5% BC (w/w), and the encapsulation efficiencies were 29.43% and 55.79%, respectively.*” (page 14, , line 257)

2. Please also provide the release behavior of Os(btpy)₂²⁺ and BODIPY-Cb from nanomicelles in physiological conditions. As the photoirradiation was conducted 24 h after the injection of NPs into mice, will the pre-release of photosensitizer and prodrug in tissue affect the photolysis as the photocleavage reaction was significantly quenched in the presence of oxygen?

Response: We agree with the review’s suggestion about testing the pre-release of photosensitizer and prodrug from micelles in physiological conditions. We here used

dialysis method to determine the release of BC and Os in 37 °C within 48 h. As shown in Figure S21, the release of BC and Os was relatively slow, which was corresponding to our previous study (*Adv. Healthcare Mater.* 9.21 (2020): 2001118). Thus, after being encapsulated in micelles, prodrugs and photosensitizer can reach tumor by EPR effect, which enables efficient light-triggered prodrug activation in tumor.

Figure S21. Release profiles of BC and Os in nanoparticles in 37 °C within 48 h.

One sentence was added to describe this additional result as: “ *Slow release of BC and Os from the nanoparticles was firstly verified (Figure S21), which is corresponding to the previously reported PLA-PEG-based prodrug nanoparticles.*” (page 16,, line 279)

3. The photolytic yields of prodrug 4-13 vary from 46% to 87%, please give the reason behind this.

Response: We appreciate the reviewer’s comment. Some possible reasons can explain the various photolytic yields of different BODIPY-based prodrugs. First, we have different linkages between the photocage and drugs, such as ester bonds (in prodrug 4-9) and carbamate bonds (in prodrug 10-13). Thus, the photolytic routes are different, which led to various photolytic yields. Besides, side reactions with the generation of byproducts reduced the yield of the free drug molecules. For example, in Figure S10 and S12, prodrug 8 and 10 generated byproducts represented by the extra peaks in HPLC traces. As a result, their photolytic yields were relatively low (48.49% and 41.34%, respectively). Usually, the side reactions are complicated and highly dependent on the drug structures (*Scientific Reports* 6.1 (2016): 21606).

Figure S10. HPLC elution curves of free IBF, BODIPY-IBF with/without NIR light irradiation in the existence of 0.1 eq Os(bptpy) $^{2+}$. (Prodrug concentration: 10^{-3} M; lamp parameter: 690 nm, 100 mW/cm 2 , 30 min; time duration: 5 min; solution: 88% methanol, 10% dichloromethane, 2% acetone, N $_2$ -saturated; $\lambda_{\text{abs}} = 254$ nm).

Figure S12. HPLC elution curves of free TCI, BODIPY-TCI with/without NIR light irradiation in the existence of 0.1 eq Os(bptpy) $^{2+}$. (Prodrug concentration: 10^{-3} M; lamp parameter: 690 nm, 100 mW/cm 2 , 30 min; time duration: 5 min; solution: 88% methanol, 10% dichloromethane, 2% acetone, N $_2$ -saturated; $\lambda_{\text{abs}} = 280$ nm).

4. There is grammar issue in line 339. Please double-check it.

Response: We acknowledge the reviewer’s comment. We have revised the corresponding content as: “Systemically administered free chlorambucil (group 2) did not exhibit detectable anti-tumor effect due to its short circulation time and rapid hydrolysis in the blood.” (page 22, , line 397)

5. For Figure 6, the order of data and its corresponding description are required to be adjusted to make them easier to follow. For example, it is better to give the prodrug activation data first.

Response: We acknowledge the reviewer’s suggestion. We have revised the Figure 6 and the corresponding descriptions in the manuscript (page 20-22).

Figure 6. (A) Representative IVIS fluorescence images of the mice after injection of free DiR and DiR NPs within 24 h (n = 3). Red dashed circles indicate tumor areas. (B)

Quantitative analysis of biodistribution in major organs and tumor determined by IVIS. Tu, He, Lu, Sp, Li, and Ki represent tumor, heart, lung, spleen, liver, and kidney, respectively. ** $p < 0.01$. (C) Schematic illustration of the treatment schedule. (D) Photograph of a mouse irradiated with NIR light at the tumor site. Concentration of (E) BC and (F) Cb in tumors and organs at 24 h after intravenous administration of Os/BC NPs with/without light irradiation on tumors ($n = 3$). (G) Photograph of tumors resected at Day 13 after different treatments. (H) Tumor volumes of each group ($n = 5$). (I) Representative H&E staining of tumor sections of different treatment groups. Scale bar: 200 μm . Light irradiation: 690 nm, 300 mW/cm^2 , 10 min. (J) Body weight of each group.

The results and discussions from Figure 6C to Figure 6I are rearranged in the order of "In vivo Imaging→Treatment Setup→Quantitative Prodrug Activation→Therapeutic Efficacy→Biological Safety". The revised contents are as follow:

“Encouraged by the anti-proliferation effect and tumor retention capability, Os/BC NPs were considered as an applicable agent for light-triggered drug release and photoactivated cancer chemotherapy. The HeLa tumor-bearing mice were randomly divided into six groups ($n = 5$) when the tumors reached at around 100 cm^3 . PBS, free Cb (8 mg/kg), Os NPs (at the equivalent concentration of Os in Os/BC NPs), BC NPs (at the equivalent concentration of Cb) and Os/BC NPs (at the equivalent concentration of Cb) were intravenously injected on Day 1 (Figure 6C). At 24 h post injection, NIR-light irradiation (690 nm, 300 mW/cm^2 , 10 min) was applied topically onto the tumor area (Figure 6D). Then LC/MS/MS was used to quantify the in vivo prodrug activation. It was found that both BC and Cb exhibited obvious relative abundance in LC/MS/MS chromatograms at low concentrations (1-400 ng/mL and 1-100 ng/mL, respectively) (Figure S25 and S26). At 24 h after the i.v. injection of OS/BC NPs, it was observed that the BC prodrug distributed mainly in tumor tissues and the major organs (Figure 6E). Obvious BC consumption was observed in tumors after NIR light irradiation, as well as the release of free Cb (Figure 6F). Notably, since the light irradiation was only performed at the tumor area, no free Cb was

observed in the other tissues, indicating the excellent tumor specificity of the treatment.

*For evaluating the therapeutic efficacy, formulation injection and light irradiation were repeated twice (Day 1 and 2; Day 6 and 7), respectively. Tumor volume was recorded within the treatment period. Obviously, Group 6 (Os/BC NPs + hv) exhibited the most obvious suppression effect on tumor growth as compared to other groups (**Figure 6G and H**). Group 3 (Os NPs + hv) displayed slightly suppression effect on tumor volume. However, no statistical difference was found between Group 3 and Groups 1, 2, 4 and 5. This result is consistent with the above finding of low cytotoxicity of Os NPs upon light irradiation, which can be explained by the limited phototoxicity of Os(btpy)₂²⁺. Systemically administered free chlorambucil (group 2) did not exhibit detectable anti-tumor effect due to its short circulation time and rapid hydrolysis in the blood.⁴⁰ On Day 13, we euthanized the mice and excised tumors and organs for ex vivo characterization. Tumor weight of Group 6 was significantly lower than other groups, demonstrating the excellent anti-tumor efficacy of Os/BC NPs with NIR light (Figure S27). Hematoxylin and eosin (H&E) staining assay was conducted to investigate the pathology of the tumors and organs. Obvious necrosis was found in the tumor tissues treated with Os/BC NPs + hv, while negligible cell apoptosis/tissue necrosis were observed in other groups (**Figure 6I**).*

*Furthermore, no obvious tissue damage was observed in major organs including heart, lung, liver, spleen, and kidney in all treated mice (Figure S28). Also, no significant change of the body weight was observed (**Figure 6J**).” (page 20-22)*

Reviewers' Comments:

Reviewer #4:

Remarks to the Author:

The authors have carefully addressed all my concerns, and I support the acceptance of this manuscript in Nature Communications now.

Reviewer #5:

Remarks to the Author:

The authors have well addressed my concerns. I thus recommend the acceptance of the manuscript by the journal as is.